# Lung adenocarcinoma cells respond differently to mechanical stress in 3D versus 2D environments
Naoya Kitamura [1] ✉, Mayumi Iwatake[2], Satoshi Mizoguchi[3], Shadil Ibrahim Wani[1], Komei Kobayashi[1,4], Muhammad Hasnain [1], Van Dung Nguyen [1], Ryo Yokoyama[1], Naru Kitade[1], Toshihiro Ojima[1], Koichiro Shimoyama[1], Naoya Koba[5], Hideki Hatta[6], Micha Sam Brickman Raredon [3], Kenichi Hirabayashi[6], Yoshitomo Morinaga[7] & Tomoshi Tsuchiya [1,4] ✉

Two-dimensional (2D) culture models are commonly used in cancer research, but fail to recapitulate complex mechanical cues of native tissues. In this study, we developed an ex vivo three-dimensional (3D) lung cancer model by seeding human lung adenocarcinoma cells into decellularised rat lungs and culturing them in a pressure chamber and perfusion system mimicking respiratory motion (RM) and blood flow. In 3D culture, RM promoted cell adhesion and proliferation, enhanced the nuclear translocation of β-catenin and YAP, and increased the expression of integrin β1 and E-cadherin. In addition, upregulation of extracellular matrix- and cell adhesion-related genes was particularly notable. In contrast, in 2D culture, RM suppressed cell proliferation and induced apoptosis, with prominent upregulation of tumour suppressor genes. Our findings demonstrate that dimensionality and mechanical stress synergistically influence lung cancer cell dynamics and underscore the need for 3D models in cancer research that closely replicate the native lung tissue microenvironment.

Cell differentiation and proliferation are influenced by the surrounding environment[1,2]. Extracellular mechanical and physical stimuli (mechanical stress), including compression, stretching, cell–cell contact, and shear stress, are converted into intracellular biochemical signals and are crucial in regulating the behaviour of normal and cancer cells[3–8]. The mechanotransduction involves complex crosstalk[9,10], and given the potential limitations of simple models in accurately reproducing such complex mechanisms[11], developing models that closely mimic the in vivo environment is essential for the study of the dynamics and behaviour of lung cancer cells.

Two-dimensional (2D) culture models have long been employed in lung cancer research with or without mechanical stresses. However, the limitations of 2D models in accurately replicating the tumour microenvironment have been widely recognised[12,13]. Cells cultured in three-dimensional (3D) structures exhibit distinct cellular morphology, tissue architecture, and protein expression compared to those cultured under 2D conditions[11,14]. Thus, the importance of 3D culture in studying

cellular dynamics is undeniable. To overcome the shortcomings of 2D models, various 3D culture techniques have been developed, including hydrogels as scaffolds, Matrigel-based organoid cultures, and microfluidic devices replicating lung structure and function, such as "Lung-on-a-chip"[15–19]. Although these models bridge the gap between 2D cell cultures and animal models[15], they are limited in fully replicating the lung's natural 3D structure, the mechanical environment induced by respiratory motion (RM), and the presence of immune cells within the tumour microenvironment[20].

The lung is a unique organ, and its 3D structure is influenced by RM (stretching stress) and blood flow (shear stress). Notably, lungs undergo periodic mechanical stresses due to respiration, exhibiting anisotropic and heterogeneous large deformations[21]. Due to its negative pressure environment and 3D scaffold, the lung differs from other parenchymal organs, such as the liver, kidneys, and heart. Therefore, an accurate understanding of lung cancer pathophysiology requires models replicating these physiological conditions.

[1]Department of Thoracic Surgery, Faculty of Medicine, University of Toyama, Toyama, Japan. [2]Research Institute for Quantum and Chemical Innovation, Institutes of Innovation for Future Society, Nagoya University, Nagoya, Japan. [3]Department of Anesthesiology, Yale University, New Haven, CT, USA. [4]Division of Cardiovascular, Respiratory and General Surgery, Program of Medical Design, Graduate School of Pharma-Medical Sciences, University of Toyama, Toyama, Japan. [5]TOKAI HIT Co., Ltd., Shizuoka, Japan. [6]Department of Diagnostic Pathology, Faculty of Medicine, University of Toyama, Toyama, Japan. [7]Department of Microbiology, Graduate School of Medicine and Pharmaceutical Sciences, University of Toyama, Toyama, Japan. ✉e-mail: naoyabacks@gmail.com; tsuchiya@med.u-toyama.ac.jp

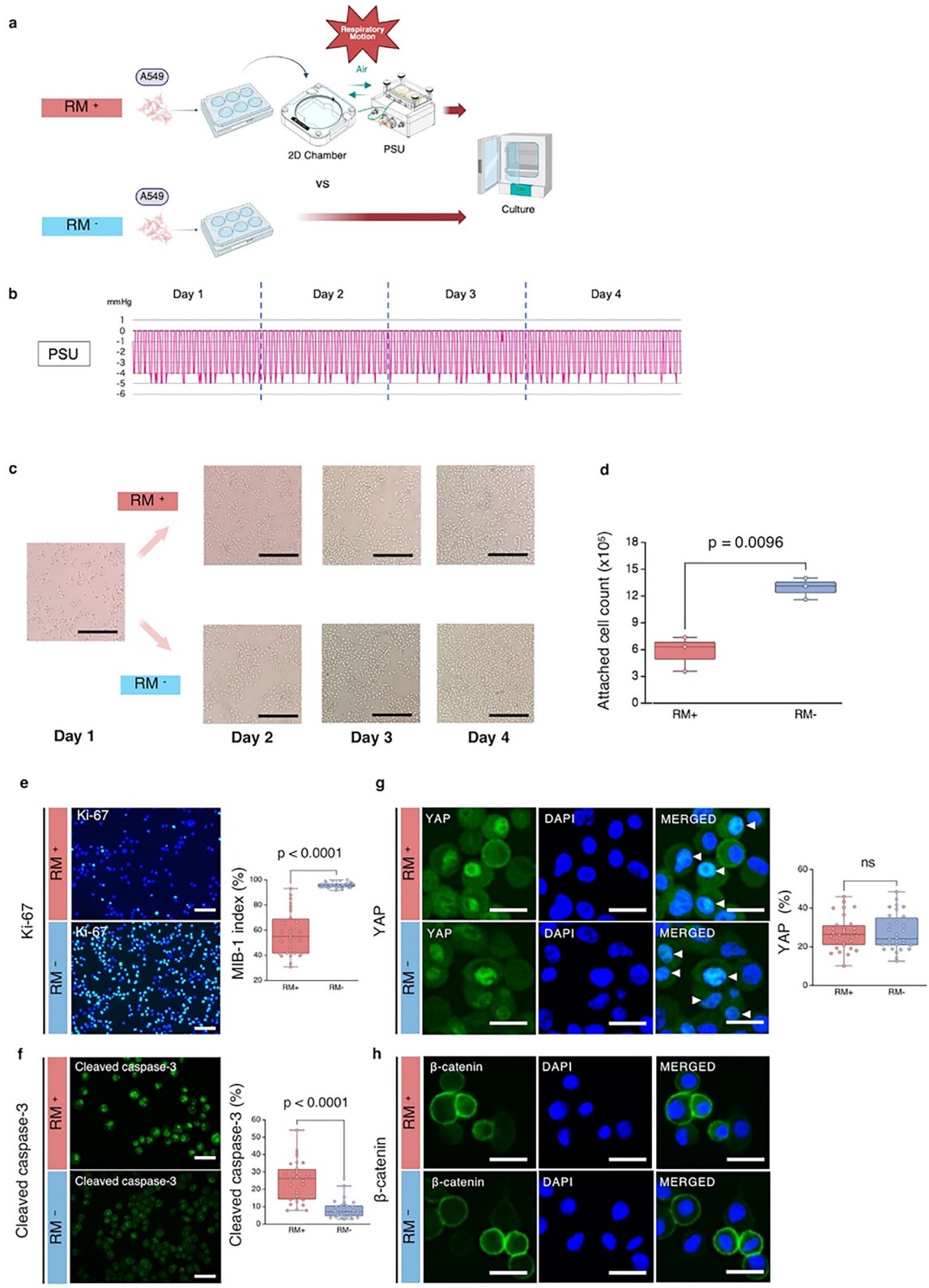

Lung decellularisation removes cellular components from tissues or organs while preserving the extracellular matrix (ECM) structure[22]. This process preserves both the alveolar and vascular scaffolds, yielding a complex ECM that functions as a 3D support structure, which has been applied in tissue engineering and regenerative medicine[23,24]. In a previous study, a 3D lung cancer model generated using decellularised rat lungs provided a more physiologically relevant cancer microenvironment than 2D cultures, enabling a detailed evaluation of cancer cell characteristics[25]. However, research on 3D lung cancer models utilising decellularisation technique is lacking, and further investigation is needed to elucidate the relationship between mechanical stress and cancer cell regulation.

**Fig. 1 | Overview and results of the 2D culture. a** Two 6-well plates were seeded with A549 cells ($3 \times 10^5$ cells/well). RM was simulated by connecting a PSU to a 2D culture chamber and applying continuous pressure changes between −5 mmHg and atmospheric pressure (0 mmHg). The RM⁺ group, in which the PSU was connected, was compared with the RM⁻ group, in which the PSU was not connected. Both plates were cultured in a 37 ℃, 5% $CO_2$ incubator ($n = 3$). Created with BioRender.com. **b** Pressure data within the RM⁺ chamber, showing continuous fluctuations between −5 mmHg and atmospheric pressure (0 mmHg) from day 1 to day 4. **c** Under microscopic examination, the RM⁻ group appeared to proliferate more rapidly. **d** Comparison of attached cell counts collected on day 4. The RM⁻ group exhibited a significantly higher cell count ($p = 0.0096$; $n = 3$). **e** Immunostaining for Ki-67. The RM⁻ group exhibited a significantly higher number of Ki-67-positive cells, and the MIB-1 index (percentage of Ki-67-positive cells) was significantly greater than in the RM⁺ group ($p < 0.0001$; $n = 3$). **f** Immunostaining for cleaved caspase-3 revealed a significantly greater number of positive cells in the RM⁺ group ($p < 0.0001$; $n = 3$). **g** Nuclear localisation of YAP was observed in both groups (white arrowheads), but no statistically significant difference was found ($p = 0.905$; $n = 3$). **h** β-catenin showed no nuclear localisation in either group ($n = 3$). Scale bars: (c) 200 μm; **e** 100 μm; **f** 50 μm; **g, h** 25 μm. Note: For (e–g), each biological sample ($n = 3$) was measured in 10 randomly selected microscopic fields (regions of interest), and all data points from these fields are shown in the scatter plots. 2D two-dimensional, RM respiratory motion, RM⁺ with RM, RM⁻ without RM, PSU pressure stimulation unit, YAP yes-associated protein, ns not significant. All error bars represent the standard deviation.

Accordingly, we established an ex vivo 3D lung cancer model by seeding human cancer cells into decellularised rat lungs and culturing them in a tightly controlled pressure chamber and perfusion system mimicking the physiological environment. Using the 3D decellularised lung cancer model, this study aimed to elucidate the impact of mechanical stress, particularly RM, on lung cancer cell dynamics. This study demonstrated that RM exerts fundamentally different effects on cancer cell behaviour in 2D and 3D systems, underscoring the importance of 3D models that replicate the physiological lung environment.

## Results

### 2D cell culture using a pressure chamber

In 2D culture, a pressure stimulation unit (PSU; TOKAIHIT, Shizuoka, Japan) served as a pressure chamber and was used to alternately change the air pressure inside a sealed chamber between −5 mmHg and atmospheric pressure (0 mmHg). This process was defined as RM. The cultures were divided into RM⁺ (with RM) and RM⁻ (without RM) groups (Fig. 1a). The pressure inside the chamber, where the well plate containing A549 cells was placed and connected to the PSU, was controlled within the range of −5 to 0 mmHg (Fig. 1b). Compared with the RM⁺ group, RM⁻ group exhibited rapid proliferation (Fig. 1c). After four days of culture, the attached cell count per well was higher in the RM⁻ group compared with the RM⁺ group ($5.74 \pm 1.97 \times 10^5$ vs $12.90 \pm 1.21 \times 10^5$, $p = 0.0096$, $n = 3$) (Fig. 1d). The number of Ki-67-positive cells (MIB-1 index), an indicator of cell proliferation, was significantly higher in the RM⁻ group ($57.43 \pm 17.39\%$ vs $95.75 \pm 1.99\%$, $p < 0.0001$, $n = 3$) (Fig. 1e). Conversely, cleaved caspase-3-positive cells, indicative of apoptosis, were significantly increased in the RM⁺ group ($25.07 \pm 11.59\%$ vs $8.07 \pm 4.27\%$, $p < 0.0001$, $n = 3$) (Fig. 1f). No significant difference was observed in nuclear staining for Yes-associated protein (YAP) ($27.27 \pm 8.77\%$ vs $27.55 \pm 9.65\%$, $p = 0.905$, $n = 3$) (Fig. 1g), and nuclear staining for β-catenin was not detected in either group (Fig. 1h).

### 3D cell culture using a pressure chamber and perfusion system

The ex vivo 3D lung cancer model was constructed by decellularising rat lungs (Supplementary Fig. 1), followed by recellularisation with A549 cells via airway seeding. The samples were cultured for four days—RM⁺ and RM⁻ conditions were compared—and subsequently harvested (Fig. 2a). A blood pressure unit (BPU; TOKAIHIT, Shizuoka, Japan) was employed for continuous vascular perfusion and pressure monitoring. Only the RM⁺ group was subjected to pressure changes inside the sealed chamber via the PSU, undergoing pressure regulation similar to that used in the 2D culture system (Fig. 2b). Following overnight preconditioning to promote peripheral vascular expansion of the decellularised lungs (i.e., continuous intravascular perfusion with culture medium at a constant flow rate), recellularisation was performed using A549 cells. As the perfusion rate increased, vascular pressure increased but was controlled to remain below an average of 20 mmHg at all times (Fig. 2c). Due to RM, the lungs inside the chamber underwent repeated expansion and contraction (Fig. 2d and Supplementary Movie 1). The baseline fluctuations in vascular pressure observed in the RM⁺ samples reflected the pressure changes inside the chamber, indicating respiratory fluctuations. The vascular pressure at each phase (day 0, 1, 2 and 3–4) tended to be lower in the RM⁺ group, presumably due to the effects of negative pressure; however, no significant difference was found between the two groups ($p = 0.72$, $0.24$, $0.45$, and $0.55$ for day 0, 1, 2, and 3–4, respectively) (Fig. 2e).

Macroscopic observations of the lungs after four days of 3D culture showed that both lungs in the RM⁺ group were extensively white, and haematoxylin and eosin (HE) staining revealed widespread cell adhesion (Fig. 3a). In contrast, in the RM⁻ group, only localised white areas were observed, primarily in the bilateral upper lobes, and HE staining showed scattered, localised cell adhesion (Fig. 3b). The cell adhesion rate was significantly higher in the RM⁺ group ($33.13 \pm 4.29\%$ vs $13.60 \pm 8.14\%$, $p = 0.01$, $n = 4$) (Fig. 3c, d). A detailed examination of HE-stained slides revealed minimal cell detachment in the RM⁺ group; widespread cytoplasmic detachment was observed in the RM⁻ group (Fig. 3e). The MIB-1 index was significantly higher in the RM⁺ group ($71.92 \pm 18.72\%$ vs $45.90 \pm 15.50\%$, $p < 0.0001$, $n = 4$) (Fig. 4a). Cleaved caspase-3 was positive only in detached cells in both groups, while adherent cells remained negative. The number of cleaved caspase-3-positive cells did not differ between the two groups ($9.77 \pm 6.59\%$ vs $10.37 \pm 9.69\%$, $p = 0.746$, $n = 4$) (Fig. 4b). The proportion of nuclei positive for YAP and β-catenin was significantly higher in the RM⁺ group than in the RM⁻ group (YAP, $2.51 \pm 2.84\%$ vs $0.46 \pm 1.11\%$, $p < 0.0001$; β-catenin, $1.78 \pm 1.55\%$ vs $0.20 \pm 0.30\%$, $p < 0.0001$; $n = 4$) (Fig. 4c, d). Furthermore, integrin β1 expression was higher in the RM⁺ group ($246.8 \pm 91.2$ vs $154.3 \pm 82.6$ per field, $p < 0.0001$, $n = 4$) (Fig. 4e), and E-cadherin fluorescence intensity was increased in RM⁺ cells ($10,167 \pm 3277$ vs $8530 \pm 2422$ per cell, $p = 0.0132$, $n = 4$) (Fig. 4f). Dual immunofluorescence staining of integrin β1 and collagen I, and DAPI counterstaining, showed evidence of cellular adhesion to the ECM mediated by integrin β1 and collagen I in the RM⁺ group, concomitant with increased integrin β1 expression (Fig. 4g). Similarly, dual immunofluorescence staining of integrin β1 and fibronectin, and DAPI counterstaining, revealed integrin β1 and fibronectin-mediated cellular adhesion to the ECM in the RM⁺ group (Fig. 4h).

### RNA sequencing and quantitative reverse transcription polymerase chain reaction (qRT-PCR)

A heatmap and volcano plot of differentially expressed genes (DEGs) ($p < 0.05$ and $\log_2$ fold change $> 1$ or $< -1$) in 2D cultures ($n = 2$) are shown (Fig. 5a). In the RM⁺ group under 2D conditions, tumour suppressor genes, *CDKN1A* and *NR4A3*, were upregulated and tumour-promoting genes, *CA9*, *EFNA1*, and *SUSD2*, were downregulated. Gene Ontology (GO) analysis revealed no prominent enrichment in any of the categories. Although these terms were not statistically significant, processes such as "negative regulation of cell growth" and "negative regulation of G1/S transition of mitotic cell cycle", which are associated with suppression of cell proliferation and cell cycle progression, were enriched in the 'Biological Process' category. In the qRT-PCR analysis, no significant differences were observed for *EFNA1*, a tumour-promoting gene, or for *integrin β1*, a well-characterised adhesion-related gene, as well as *CTGF* and *CCND1* (encoding cyclin D1), both of which are upregulated by YAP nuclear translocation when the Hippo pathway is inactivated. In contrast, the

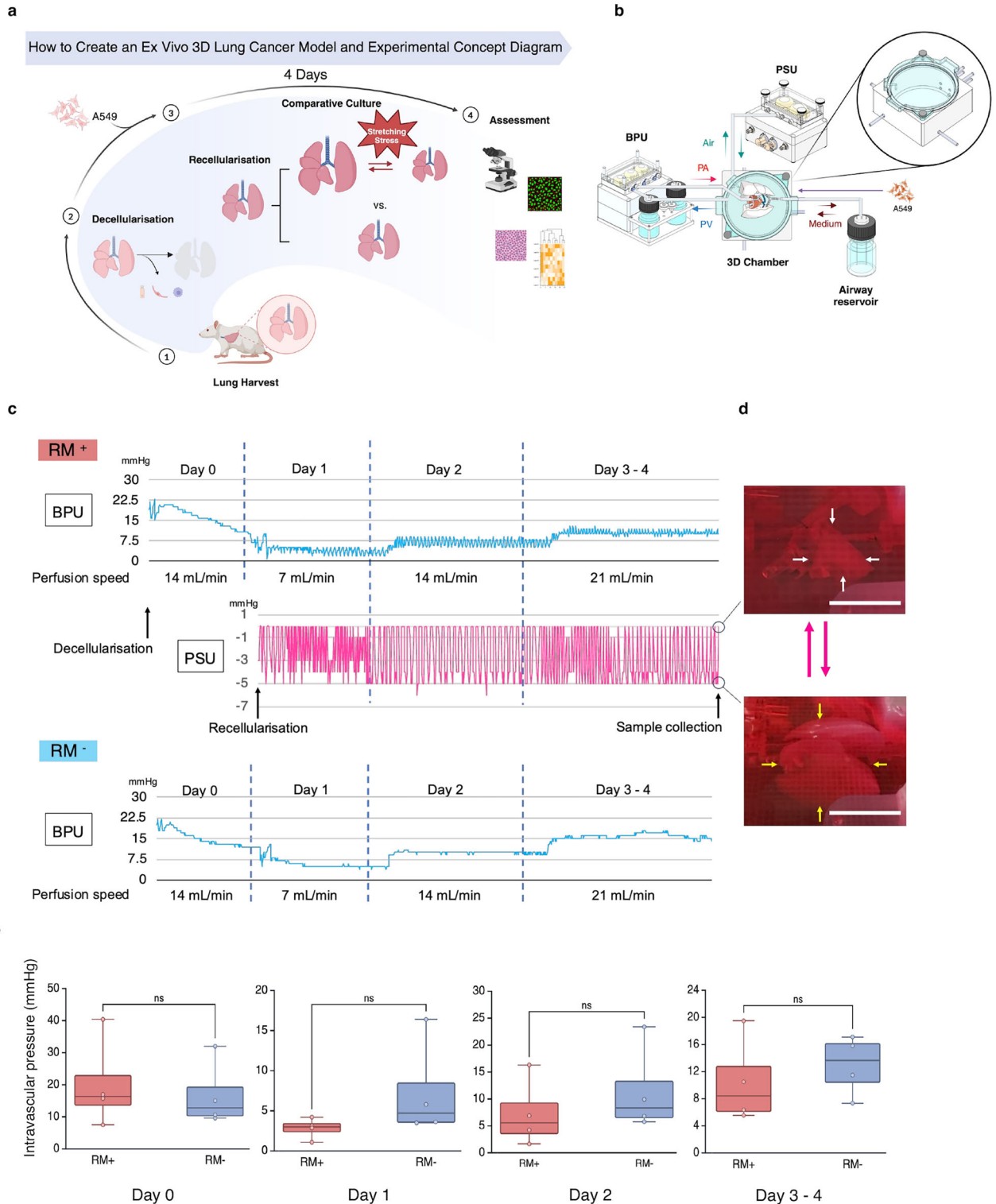

tumour suppressor genes *CDKN1A* (p = 0.0343) and *NR4A3* (p = 0.0493) were significantly upregulated in the RM⁺ group (Fig. 5b).

In 3D cultures (*n* = 4), *INHBA, LOX, LRRC15*, and *CXCL12*, potentially involved in cell proliferation and adhesion, were upregulated (Fig. 6a). GO analysis showed no statistically significant enrichment in the 'Biological Process' category. However, "extracellular space", "protein complex involved in cell–cell adhesion", and "extracellular region" under the

'Cellular Component' category, were enriched. "Cytokine activity" was significantly enriched under the 'Molecular Function' category. In the qRT-PCR analysis, no statistically significant differences were observed for *LOX, CXCL12*, or *CCND1*. By contrast, *integrin β1* (p = 0.0277), which is involved in cell adhesion, and *CTGF* (p = 0.0296), a canonical YAP target gene upregulated upon YAP nuclear translocation, were significantly upregulated in the RM⁺ group. (Fig. 6b). Gene set enrichment analysis (GSEA) identified

**Fig. 2 | Construction of the ex vivo 3D lung cancer model and overview of the pressure chamber and perfusion system. a** Lungs and heart were harvested *en bloc* from rats, and decellularisation was performed by perfusing the airway and vasculature with solutions containing Triton X-100, Benzonase, and sodium deoxycholate. A549 cells were subsequently seeded *via* the airway and allowed to adhere (recellularisation) to create an ex vivo 3D lung cancer model. The samples were cultured for four days under RM (RM⁺) or without RM (RM⁻), followed by analysis. Created with BioRender.com. **b** Overview of the pressure chamber and perfusion system. The decellularised lung was placed in a chamber filled with culture medium. The airway was connected to an airway reservoir, the PA to the BPU, and the cannula in the PV (left ventricle) was left open to the medium inside the chamber. This configuration enabled passive medium exchange *via* the airway, and a perfusion circuit was established from PA through the lung to the PV and back to the BPU. The chamber was sealed and connected to a PSU, which continuously modulated the intrachamber pressure to simulate a negative pressure environment and RM, mimicking the thoracic cavity. **c** Monitoring data during 3D culture in the RM⁺ and

RM⁻ groups. On the evening of day 0, intravascular perfusion (14 mL/min) was initiated to promote peripheral vascular expansion in the lung. The perfusion rate was increased every 24 h (7 → 14 → 21 mL/min). Only the RM⁺ group was subjected to continuous pressure fluctuations between –5 mmHg and atmospheric pressure from the time of recellularisation on day 1. The vascular pressure waveform showed fine oscillations reflecting respiratory variation once the PSU was activated. **d** Representative images of the lung inside the chamber. The lung expands under negative pressure (yellow arrows) and contracts at atmospheric pressure (white arrows). Scale bars: 2 cm. **e** Comparison of vascular pressures between the RM⁺ and RM⁻ groups throughout the procedure revealed no statistically significant differences. The final vascular pressures on days 3–4 were $10.47 \pm 6.40$ mmHg and $12.93 \pm 4.44$ mmHg, respectively. 3D three-dimensional, RM respiratory motion, RM⁺ with RM, RM⁻ without RM, PA pulmonary artery, BPU blood pressure unit, PV pulmonary vein, PSU pressure stimulation unit, ns not significant. All error bars represent the standard deviation.

one significantly enriched gene set in the RM⁺ group (Androgen Response; enrichment score (ES): 0.3, normalized enrichment score (NES): 1.44, $p < 0.0001$) and three in the RM⁻ group (Peroxisome; ES: $-0.33$, NES: $-1.59$, $p < 0.0001$; Acid Metabolism; ES: $-0.41$, NES: $-1.58$, $p < 0.0001$; P53 Pathway; ES: $-0.28$, NES: $-1.42$, $p = 0.0059$) (Fig. 6c). DEGs upregulated or downregulated in the 2D and 3D models were 66/31 and 63/64, respectively. Among them, *SIK1* was the only commonly regulated DEG in both models (Fig. 6d).

## Discussion

In the ex vivo 3D lung cancer model, the group subjected to RM—mimicking human-like breathing—exhibited enhanced cell adhesion and proliferation compared to the group with no motion, accompanied by changes in gene expression and signalling pathways that could explain these phenomena. In contrast, adding RM in 2D cultures led to suppressed cell proliferation and a markedly different gene expression profile in RNA sequencing compared to the 3D model. These findings highlight that cellular behaviour is strongly influenced by the experimental model and external mechanical environment. Our findings emphasise the importance of incorporating RM in a 3D ECM that supports mechanosensing to investigate lung cancer biology more accurately.

Lung adenocarcinoma cells are naturally exposed to continuous mechanical stimuli within the pulmonary microenvironment, including pressure fluctuations, stretch forces associated with RM, and blood flow. A bioreactor, as a closed system, is capable of controlling environmental parameters such as temperature, oxygen, pressure, and nutrient supply to maintain and promote biological processes under optimal conditions. Bioreactors incorporating combined perfusion and ventilation models have been reported to improve nutrient distribution and promote cell proliferation, thereby resulting in more uniform cell distribution[5,26,27]. In our study, a bioreactor equipped with a pressure chamber and perfusion system successfully provided an environment in which both vascular and intrathoracic pressures remained within the physiological ranges reported in humans[28–30]. Notably, unlike conventional volume-controlled systems[25], our system employed pressure-controlled regulation, enabling the reproduction of stretch stress generated by large, multidirectional deformation of lung tissue[21].

Taken together, these findings indicate that the bioreactor used in this study provides a platform that closely replicates the physiological lung microenvironment and offers a robust system for investigating the mechanobiological responses of lung cancer cells.

In the 2D RM⁺ group, cell proliferation was suppressed, and apoptosis was induced. RNA sequencing revealed upregulation of tumour suppressor genes, including *CDKN1A* and *NR4A3*[31,32], and downregulation of tumour-promoting genes, including *CA9*, *EFNA1*, and *SUSD2*[33–36]. Furthermore, nuclear translocation of YAP was observed in both RM⁺ and RM⁻ groups, with no significant difference between the two conditions, whereas β-catenin did not exhibit nuclear translocation in either group. qRT-PCR

analysis also confirmed significant upregulation of tumour suppressor genes, supporting the RNA-seq findings. Collectively, these results suggest that in a 2D environment, where only a rigid substrate is provided and the native softness of the tissue is not recapitulated, negative pressure functions as a stressor, and RM alone is insufficient to activate intracellular signalling pathways.

In contrast, in the 3D RM⁺ group, apoptosis was not induced; instead, the proliferation rate increased, accompanied by a marked enhancement of cell adhesion. RNA sequencing demonstrated upregulation of genes involved in cell adhesion and proliferation, such as *INHBA*, *LOX*, *LRRC15*, and *CXCL12*[37–41], and GO analysis revealed significant enrichment of genes related to cell–cell adhesion and ECM components. GSEA revealed an enriched gene set of the P53 pathway in the RM⁻ group, suggesting that, unlike in the 2D environment, the absence of RM may act as a stressor for cells. Moreover, qRT-PCR confirmed upregulation of *integrin β1* and *CTGF*, which was not observed in the 2D environment. Taken together, these results indicate that in the 3D model, in which the alveolar scaffold functions as a supportive structure for attaching cells, negative pressure does not act as a stressor; rather, mechanosensing and activation of the YAP/TAZ pathway tend to predominate. Therefore, the increased proliferation observed in the 3D RM⁺ group is more likely attributable not to the direct effect of RM itself but to the secondary consequence of enhanced stability of cell adhesion induced by RM.

These findings underscore that even when the same cell line is exposed to identical RM, gene expression responses differ substantially depending on dimensionality and the physical scaffolding provided. In other words, the "input" of mechanical stimulation (rigid substrate versus compliant alveolar scaffold) and the "output" (cell-cycle regulation versus ECM remodelling) may be switched according to dimensional context. Indeed, only one DEG, *SIK1*, was shared between the 2D and 3D models, supporting this interpretation.

Previous studies have demonstrated that 3D models more accurately recapitulate the physiological dynamics of cancer cells[14,42–46]; however, few have directly compared dimensional models under mechanical stress. In this study, we performed such a direct comparison and confirmed that 3D models faithfully reproduce both the lung tissue microenvironment and the mechanical cues present in vivo, thereby highlighting the profound influence of model choice on cellular behaviour.

Integrin β1 is a well-characterised mechanosensor that mediates cell adhesion by binding to components such as fibronectin, collagen, and laminin, thereby converting mechanical stimuli into intracellular biochemical responses[47–49]. E-cadherin plays a critical role in cell–cell adhesion and epithelial–mesenchymal transition[50,51]. In this study, the RM⁺ group showed a tendency toward co-localisation of integrin β1 with fibronectin or collagen I, along with enhanced expression of E-cadherin, whereas these features were insufficient in the RM⁻ group. These observations suggest that RM-induced three-dimensional expansion of the lung[21,52] enhances mechanosensitivity through alveolar scaffold stretching, thereby promoting

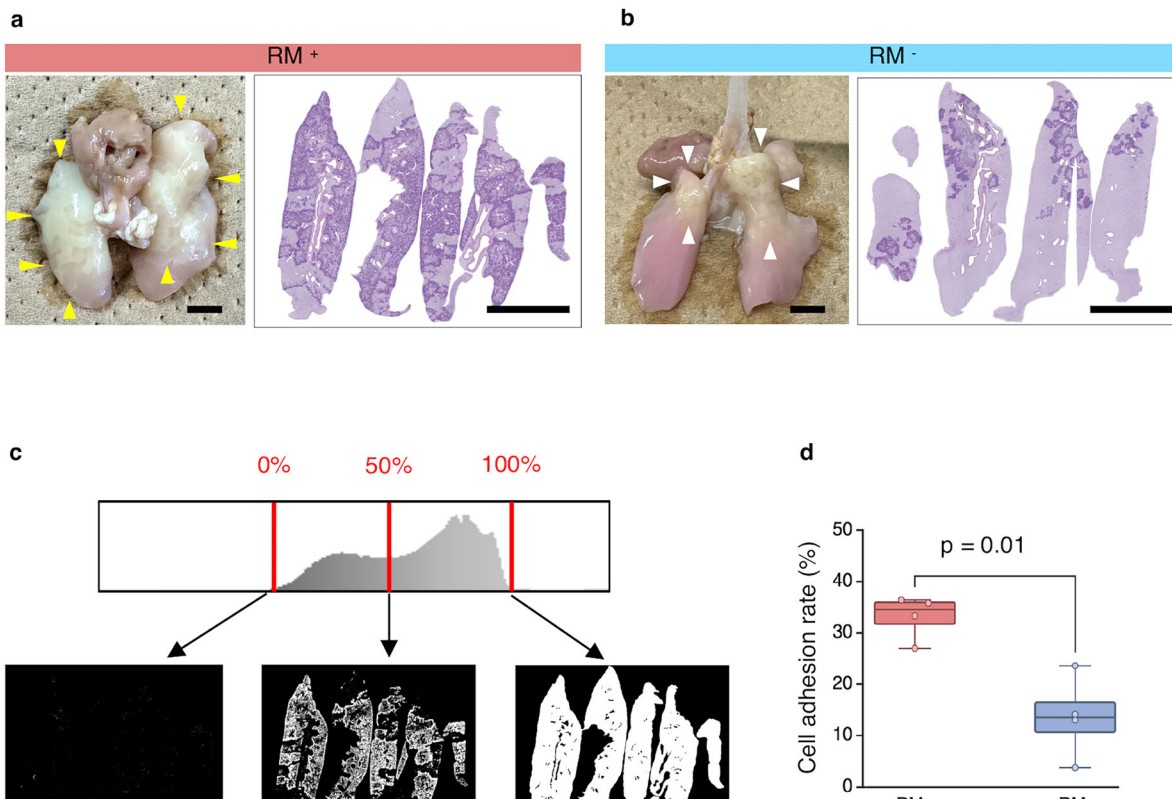

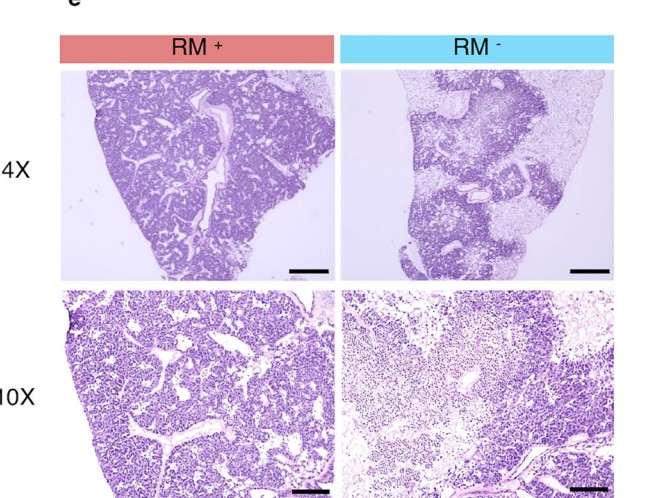

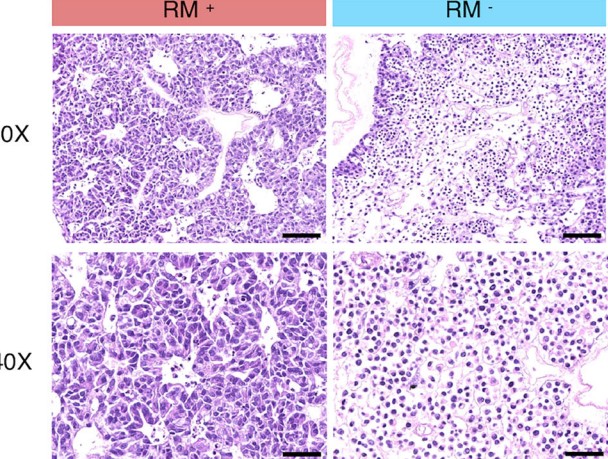

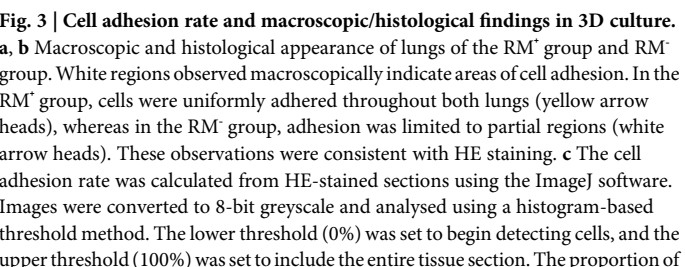

**Fig. 3 | Cell adhesion rate and macroscopic/histological findings in 3D culture.** **a, b** Macroscopic and histological appearance of lungs of the RM⁺ group and RM⁻ group. White regions observed macroscopically indicate areas of cell adhesion. In the RM⁺ group, cells were uniformly adhered throughout both lungs (yellow arrow heads), whereas in the RM⁻ group, adhesion was limited to partial regions (white arrow heads). These observations were consistent with HE staining. **c** The cell adhesion rate was calculated from HE-stained sections using the ImageJ software. Images were converted to 8-bit greyscale and analysed using a histogram-based threshold method. The lower threshold (0%) was set to begin detecting cells, and the upper threshold (100%) was set to include the entire tissue section. The proportion of the area detected at the 50% threshold was defined as the cell adhesion rate: Cell adhesion rate (%) = (Area at 50% threshold / Area at 100% threshold) × 100. **d** Comparison of cell adhesion rates. The RM⁺ group showed a significantly higher adhesion rate than the RM⁻ group (p = 0.01; $n = 4$). **e** High-power views of HE-stained sections. In the RM⁺ group, cells were widely and uniformly adhered with minimal detachment, whereas in the RM⁻ group, cell adhesion was sparse and irregular, showing frequent areas of detachment. Scale bars: **a, b** 5 mm; **e** (4×) 500 μm; (10×) 200 μm; (20×) 100 μm; (40×) 50 μm. 3D three-dimensional, RM respiratory motion, RM⁺ with RM, RM⁻ without RM, HE haematoxylin and eosin. All error bars represent the standard deviation.

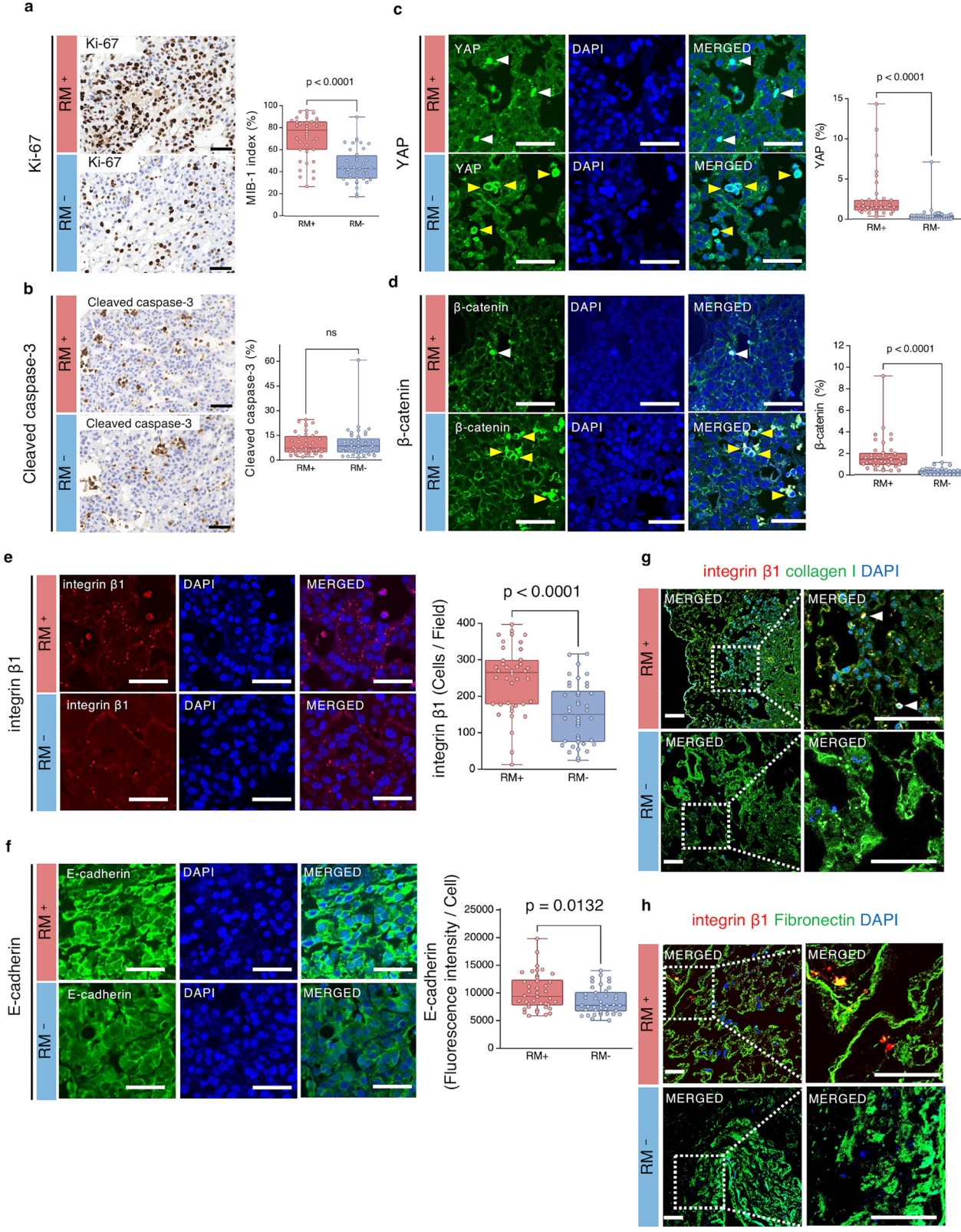

the expression of adhesion-related proteins that strengthen both cell–scaffold and cell–cell interactions.

Moreover, increased nuclear translocation of β-catenin and YAP, together with upregulation of the downstream factor CTGF, was observed, indicating that the process by which cells sense and transmit physical stress was facilitated. β-catenin is involved in both cell adhesion and Wnt signalling; upon nuclear translocation, it binds to transcription factors (TCF/LEF) to promote gene expression[47]. Cyclic stretch has also been reported to enhance Wnt/β-catenin signalling in A549 cells[50].

Taken together, these findings indicate that RM-induced signal transduction first involves the upregulation of integrin β1 and the enhancement of cellular adhesion. Signals are then transmitted, either

**Fig. 4 | Immunohistochemistry and immunofluorescence in 3D culture. a** Ki-67-positive cells were more frequently observed in the RM⁺ group, and the MIB-1 index (percentage of Ki-67-positive cells) was significantly higher than that in the RM⁻ group (p < 0.0001; *n* = 4). **b** Cleaved caspase-3 positivity was detected only in detached cells in both groups, with no statistically significant difference (*n* = 4). **c, d** The proportion of cells showing nuclear localisation of YAP (p < 0.0001) and β-catenin (p < 0.0001) was significantly higher in the RM⁺ group (white arrowheads) (*n* = 4). In contrast, in the RM⁻ group, cytoplasmic staining without nuclear localisation was frequently observed (yellow arrowheads). **e** Integrin β1 expression was significantly higher in the RM⁺ group (p < 0.0001; *n* = 4). **f** Fluorescence intensity of

E-cadherin was enhanced in the RM⁺ group (p = 0.0132; *n* = 4). **g** In the RM⁺ group, findings suggestive of cellular adhesion to the ECM mediated by integrin β1 and collagen I were observed (white arrowheads). **h** In the RM⁺ group, cellular adhesion to the ECM mediated by integrin β1 and fibronectin was observed. Scale bars: (**a–h**) 50 μm. Note: For (**a–f**), each biological sample (*n* = 4) was measured in 10 randomly selected microscopic fields (regions of interest), and all data points from these fields are displayed in the scatter plots. 3D three-dimensional, RM respiratory motion, RM⁺ with RM, RM⁻ without RM, YAP yes-associated protein, ECM extracellular matrix, ns not significant. All error bars represent the standard deviation.

directly via integrin β1 to β-catenin and YAP[53] or indirectly, culminating in the upregulation of *CTGF*, which contributes to functional responses such as enhanced proliferation and adhesion[54]. Accurate replication of mechanical stretch is therefore essential for understanding cellular responses[51], and our study underscores the necessity of incorporating RM in 3D models to investigate lung cancer biology with physiological fidelity.

Currently, experimental approaches to reproduce 3D environments include hydrogel-based culture systems and microfluidic technologies such as the "Lung-on-a-chip" model[15–18]. Each of these models possesses distinct advantages and limitations and differs substantially from the ex vivo 3D lung cancer model employed in this study.

Hydrogels enable 3D cell culture and provide a more physiologically relevant context for cell–cell and cell–ECM interactions compared with 2D culture[16]. However, the composition of Matrigel differs from that found in the tumour microenvironment (e.g., complexes containing structural proteins such as collagen and elastin), making it difficult to faithfully reproduce ECM components derived from specific tissues or cell types.

The "Lung-on-a-chip" system utilises microfluidic devices to culture cells within microscale channels that mimic the dynamic environment of the lung[55]. A notable advantage of this model is the ability to establish air–liquid interface cultures, which are essential for replicating lung-specific functionality. Nevertheless, the complexity of device fabrication and operation, as well as limitations in culture scale, make this model less suitable for tissue-level investigations[56].

In contrast, our ex vivo 3D model preserves the microenvironment of actual lung tissue, enabling observation of cellular adhesion, proliferation, and tissue-level responses. This model allows investigation of changes in signalling pathways dependent on the ECM, which functions as a structural and biochemical scaffold—features that are difficult to reproduce using conventional 2D culture or artificial matrices such as hydrogels. However, in the present study, we did not directly compare our model with other widely used 3D systems, such as organoids or "Lung-on-a-chip," and therefore its relative superiority within the broader landscape of 3D culture systems remains to be determined.

The DEGs identified in this study were based on p < 0.05 without multiple testing correction and should therefore be interpreted cautiously, acknowledging their exploratory nature. The incomplete concordance between RNA-seq and qRT-PCR results may reflect differences in statistical thresholds and sample sizes, as well as the influence of post-transcriptional regulation or mRNA stability. Accordingly, in interpreting the results, we placed greater emphasis on the consistency of directional changes rather than on absolute values.

This study revealed substantial differences in gene and protein expression between 2D and 3D cultures, as well as between conditions with and without RM. These findings demonstrate that cellular behaviour and gene expression on 2D substrates do not fully recapitulate those observed in 3D environments or in vivo. Our work highlights the limitations of conventional 2D models in basic research and emphasises the need for 3D culture systems that incorporate RM and appropriate scaffolding. While the insights obtained here are based on in vitro models and should be extrapolated to clinical contexts with caution, the observation that the mechanical environment encountered by tumours may govern signalling pathways has potential implications for clinical practice. Specifically, it may inform strategies related to the control of

intrathoracic pressure and ventilation and may inspire novel therapeutic approaches targeting the tumour microenvironment. Future studies should aim to validate causal relationships using approaches such as integrin β1–blocking antibodies, YAP inhibitors, and traction force microscopy, and additional investigations are warranted to account for factors such as oxygen and nutrient diffusion gradients and cell density effects in 3D cultures.

## Methods
### Harvest of rat heart–lung blocks
Lungs were obtained from 6- to 10-week-old male Sprague–Dawley (SD) rats (Jackson Laboratory Japan, Inc., Kanagawa, Japan). All animal procedures were approved by the Animal Care and Use Committee of the University of Toyama (approval number: A2023UH-01) and conducted in accordance with the *Guide for the Care and Use of Laboratory Animals*. We have complied with all relevant ethical regulations for animal use.

Rats were anaesthetised with inhaled isoflurane (DS Pharma Animal Health, Osaka, Japan), followed by intraperitoneal administration of a mixed anaesthetic cocktail (total volume: 25 mL) comprising 1.5 mL of medetomidine hydrochloride (1 mg/mL; Nippon Zenyaku Kogyo Co., Ltd., Fukushima, Japan), 4 mL of midazolam (5 mg/mL; Sandoz K.K., Tokyo, Japan), 5 mL of butorphanol tartrate (5 mg/mL; Meiji Animal Health Co., Ltd., Tokyo, Japan), and 14.5 mL of saline. The dose was adjusted to 0.5 mL per 200 g of body weight to ensure adequate analgesia and sedation.

A tracheotomy was performed, and a 16 G catheter was inserted into the trachea for intubation. The rats were on a ventilator at a tidal volume of 10 mL/kg with a respiratory rate of 90 breaths per min. After a transverse abdominal incision, the anterior thoracic wall was removed. Anticoagulation was achieved by injecting heparin sodium (1000 U/kg; Mochida Pharmaceutical Co., Ltd., Tokyo, Japan) via the inferior vena cava, followed by transection of the cardiac apex.

A 16 G catheter was inserted into the pulmonary artery via the right ventricle, and the lungs were perfused with 50 mL of phosphate-buffered saline (PBS) containing heparin sodium (50 U/mL) and sodium nitroprusside dihydrate (Sigma-Aldrich, St. Louis, MO, USA; 10 μg/mL) to flush out the blood. The lungs were excised en bloc together with the heart and trachea. These procedures were performed approximately once or twice per month.

### Decellularisation of the lung tissue
Cannulas were inserted into the trachea, pulmonary artery, and left ventricle (pulmonary vein), and lungs were decellularised following a previously reported method[25,57]. In brief, the pulmonary vasculature was perfused *via* the pulmonary artery with PBS containing calcium and magnesium (PBS⁺) supplemented with heparin sodium, sodium nitroprusside, antibiotics, and 0.0035% Triton X-100 (Nacalai Tesque, Kyoto, Japan). Subsequently, 20 units of Benzonase (25 U/μL; Enzynomics, Daejeon, South Korea) in Benz buffer (50 mM Tris-HCl, 0.1 mg/mL BSA, 1 mM MgCl₂) was administered via the airway.

Next, sodium deoxycholate (SDC; Nacalai Tesque, Kyoto, Japan) at concentrations of 0.01%, 0.05%, and 0.1% in PBS without calcium and magnesium (PBS⁻) was perfused through the vasculature, followed by

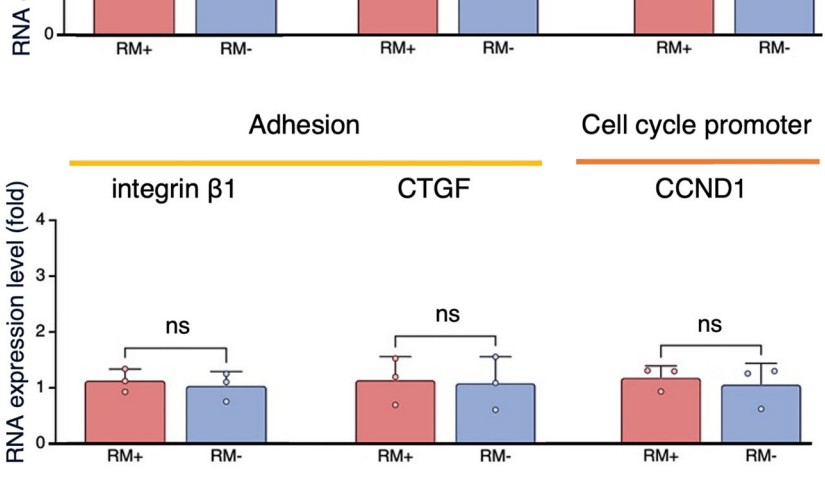

airway administration of 20 units of Benzonase in Benz buffer. After vascular perfusion with 0.5% Triton X-100, the lungs were flushed with PBS⁻ and perfused with PBS⁻ containing antibiotics (penicillin–streptomycin, amphotericin B, and gentamicin) (Supplementary Fig. 1a). Decellularised lungs were temporarily stored in PBS⁻ containing antibiotics (Supplementary Fig. 1b).

## Culture and preparation of human lung cancer cells

The human lung adenocarcinoma cell line A549 was obtained from the Japanese Collection of Research Bioresources (Osaka, Japan; https://cellbank.nibiohn.go.jp). Cells were cultured in Dulbecco's modified Eagle's medium (DMEM, high glucose, 4.5 g/L; Nacalai Tesque, Kyoto, Japan), supplemented with 10% foetal bovine serum (FBS; Sigma-Aldrich,

**Fig. 5 | RNA sequencing and qRT-PCR validation in 2D culture. a** Heatmaps and volcano plots of DEGs (non-adjusted p < 0.05 and log$_2$ fold change > 1 or < −1) between the RM⁺ group and RM⁻ group in 2D culture conditions (n = 2). Results of GO analysis are shown. In the 2D culture, the RM⁺ group showed upregulation of tumour suppressor genes, such as *CDKN1A* and *NR4A3*, and downregulation of tumour-promoting genes, including *CA9, EFNA1*, and *SUSD2*. No significant enrichment was observed in GO analysis. Although not statistically significant, enrichment trends were found under the Biological Process category for terms including "negative regulation of cell growth" and "negative regulation of G1/S transition of mitotic cell cycle". **b** In the qRT-PCR analysis, no significant differences were observed for *EFNA1, integrin β1, CTGF*, or *CCND1*. The tumour suppressor genes *CDKN1A* (p = 0.0343) and *NR4A3* (p = 0.0493) exhibited significantly higher expression in the RM⁺ group. Data are presented as mean ± SD (n = 3). Error bars indicate the SD from three independent experiments. qRT-PCR quantitative reverse transcription polymerase chain reaction, DEG differentially expressed gene, RM respiratory motion, RM⁺ with RM, RM⁻ without RM, 2D two-dimensional, GO gene ontology, SD standard deviation, ns not significant. All error bars represent the standard deviation.

St. Louis, MO, USA), 1% penicillin–streptomycin (10,000 U/mL penicillin and 10 mg/mL streptomycin; Nacalai Tesque, Kyoto, Japan), 1% amphotericin B (250 µg/mL; FUJIFILM, Tokyo, Japan), and 0.5% gentamicin (10 mg/mL; Gibco, Waltham, MA, USA). Cells were maintained at 37 °C in a humidified atmosphere containing 5% CO$_2$. For passaging or harvesting, cells were detached using 0.05% trypsin–EDTA solution (Nacalai Tesque, Kyoto, Japan) at 37 °C for 5–7 min.

### Pressure chamber and perfusion system setup for the 3D culture
Decellularised lungs with cannulas in place were placed into the chamber, with only the trachea and pulmonary artery connected to the tubing. The cannula inserted into the left ventricle (representing the pulmonary vein) was left open to the culture medium inside the chamber. The medium collected from the pulmonary vein side—from within the chamber—was circulated into the pulmonary artery via the BPU, enabling continuous intravascular perfusion and pressure monitoring (Fig. 2b).

The tubing connected to the airway was linked to a reservoir bottle containing 60 mL of culture medium, allowing for passive medium movement in response to changes in lung volume. The chamber was connected to the PSU, which continuously regulated and monitored air pressure to maintain the desired pressure settings (Fig. 2b). Intrachamber pressure measurements were taken once every 300 s (Fig. 2c).

### Pressure chamber setup for the 2D culture
A549 cells were seeded onto plastic well plates and cultured in a conventional 2D environment to evaluate the effect of RM—represented by pressure fluctuations within the chamber. Two 6-well plates (SPL Life Sciences, Pocheon, South Korea) were prepared, with 3 × 10$^5$ cells seeded per well. One plate was placed in a 2D culture pressure chamber and connected to the PSU, referred to as the RM⁺ group, while the other plate was not connected to the PSU and served as the RM⁻ group (Fig. 1a).

Both plates were incubated under standard conditions (37 °C, 5% CO$_2$) for four days. After incubation, cells were collected for further analysis. Intrachamber pressure measurements were taken once every 300 s (Fig. 1b).

### Seeding and perfusion culture of lung cancer cells
Two decellularised lungs and corresponding chambers were prepared to compare conditions with and without RM. On the evening of day 0, vascular preconditioning of the decellularised lungs was initiated to promote peripheral vascular expansion prior to cell seeding. Specifically, each lung was placed in a chamber filled with 200 mL of DMEM and connected to vascular perfusion at a flow rate of 14 mL/min in an incubator at 37 °C with 5% CO$_2$. This procedure was intended to prevent inadequate perfusion caused by vasoconstriction following decellularisation and was unrelated to either the decellularisation process or any cellular intervention. During this period, neither chamber was subjected to RM.

On day 1, A549 cells were seeded (recellularised) *via* the airway. While applying a negative pressure of −10 mmHg using the PSU to expand the lung in the sealed chamber, 4–5 × 10$^7$ cells suspended in 10 mL of DMEM were injected through the airway using a syringe. To prevent the backflow of cells, the airway tubing was clamped, and the lungs were kept stationary for 1 h in an incubator at 37 °C with 5% CO$_2$.

Subsequently, vascular perfusion was resumed in both chambers at a flow rate of 7 mL/min, and RM was initiated in only one chamber (RM⁺ group) while the other remained static (RM⁻ group). RM was simulated by cyclically modulating the intrachamber pressure between −5 mmHg and atmospheric pressure using the PSU, inducing expansion and contraction of the lung tissue, respectively (Fig. 2d). The vascular perfusion rate was gradually increased every 24 h: 7 mL/min → 14 mL/min → 21 mL/min. After 72 h of culture, the lungs were harvested for further analysis. Four independent sets of RM⁺ and RM⁻ groups were prepared for comparative analysis.

The mean pulmonary arterial pressure in healthy individuals has been reported to be ~14.0 ± 3.3 mmHg, with little variation across sex or ethnicity[28]. According to current diagnostic criteria, pulmonary hypertension is defined as a mean pulmonary arterial pressure exceeding 20 mmHg[29]. Therefore, in this model, perfusion was set such that intravascular pressures remained below 20 mmHg, allowing reproduction of the vascular shear stress generated by blood flow in vivo. The negative pressure used to mimic RM was determined based on reported intrathoracic pressures in humans. Intrapleural pressure averages approximately −5 cmH$_2$O (≈ −3.7 mmHg) and reaches about −7.5 cmH$_2$O (≈ −5.5 mmHg) during inspiration[58]. Direct measurements in healthy individuals have also reported end-expiratory pressures of approximately −2.5 to −3 mmHg[30]. Based on these findings, we applied cyclic negative pressure ranging from 0 to −5 mmHg to approximate the intrathoracic pressure fluctuations observed during spontaneous breathing.

### Rationale for the time point for evaluation
The time point for evaluation was set at 72 h. Transcriptional responses to mechanical stimulation typically emerge within 24–48 h[59,60], while processes such as cell adhesion and tissue organization generally require several days[57,61]. Thus, 72 h allowed assessment at a time when early transcriptional responses had stabilized and adhesion processes were underway, while also avoiding the decline in cell viability observed in cultures exceeding 1 week.

### Histology, immunohistochemistry, and immunofluorescence
For 2D cultures, cells were collected using 0.05% trypsin–EDTA solution and put into Millicell EZ SLIDE 8-well glass slides (Merck KGaA, Darmstadt, Germany) for subsequent staining.

For 3D cultures, the entire left lung was fixed in 10% neutral buffered formalin (Nacalai Tesque, Kyoto, Japan), sliced longitudinally into 4–5 sections, embedded in paraffin, and sectioned at a thickness of 5 µm. HE staining was performed following standard protocols.

For immunohistochemistry, paraffin sections were deparaffinised with xylene and ethanol, followed by washing with PBS. Antigen retrieval was performed in 10 mM citrate buffer (pH 6.0) at temperatures exceeding 100 °C for 10 min. The membrane was permeabilised using 0.1% Triton X-100 for 15 min, and non-specific binding was blocked with Blocking One Histo (Nacalai Tesque, Kyoto, Japan) for 15 min. Primary antibodies were diluted in SignalStain Antibody Diluent (Cell Signaling Technology, Danvers, MA, USA), and samples were incubated overnight at 4 °C. After PBS washing, samples were incubated with secondary antibodies at room temperature (approximately 21 °C) for 1 h and mounted using VECTASHIELD mounting medium with DAPI (Vector Laboratories, Inc., Hercules, CA, USA).

The primary antibodies used in this study included anti-Ki-67 (418071; Nichirei Biosciences, Tokyo, Japan), cleaved caspase-3 (25128-1-AP; Proteintech, IL, USA), β-catenin (84805; Cell Signaling Technology, MA, USA),

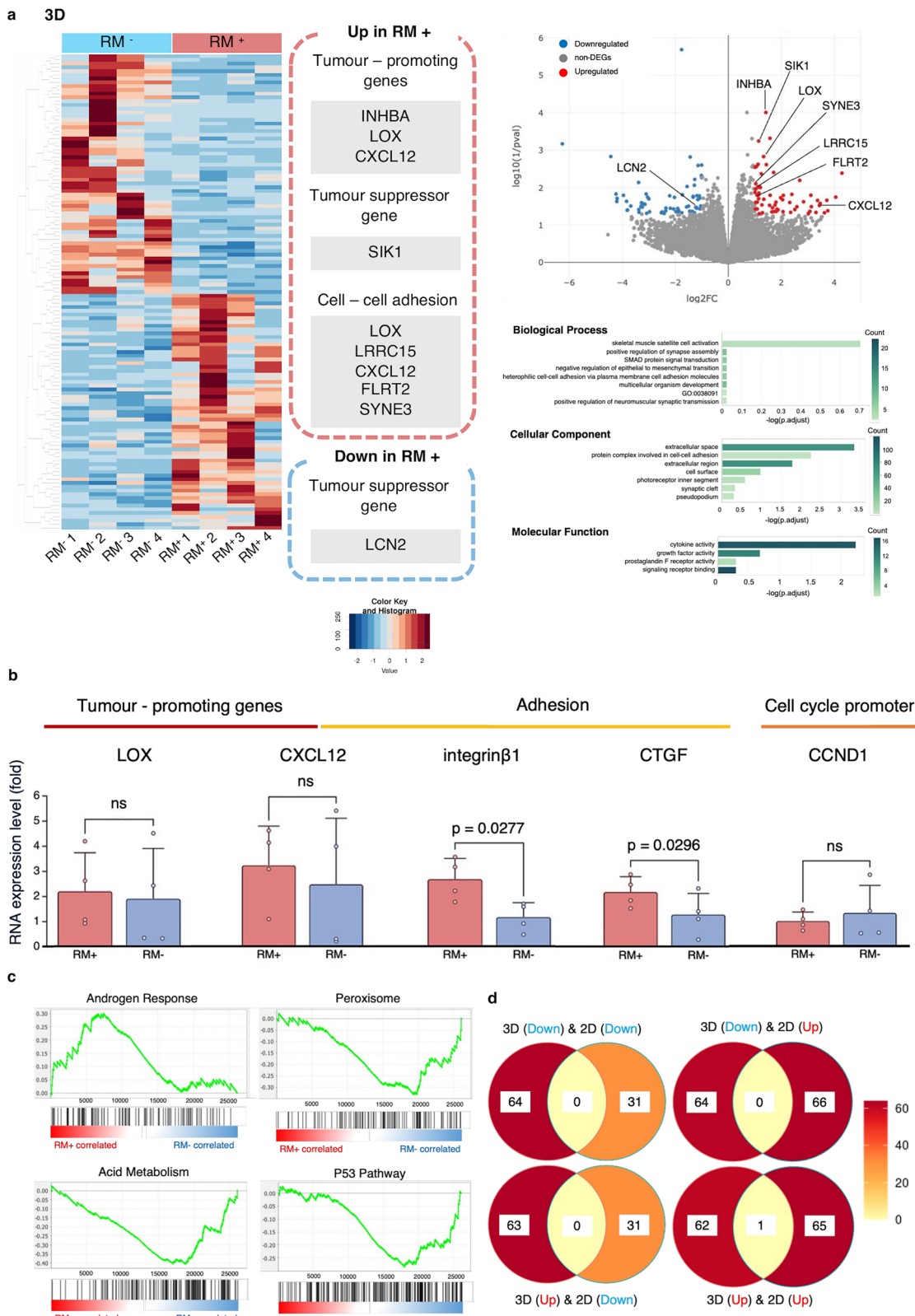

integrin β1 (ab30394; Abcam, Cambridge, United Kingdom), and E-cadherin (20874-1-AP; Proteintech, IL, USA), fibronectin (ab314679; Abcam, Cambridge, United Kingdom), and type I collagen (28368; Cell Signaling Technology, MA, USA). Secondary antibodies conjugated with Alexa Fluor 488 or Alexa Fluor 555 (Thermo Fisher Scientific, MA, USA) were used as appropriate.

**Image acquisition and intensity normalisation for quantification**

Images were acquired using a fluorescence microscope (BZ-X800, Keyence, Osaka, Japan) or a confocal laser scanning microscope (LSM780, Carl Zeiss, Jena, Germany). Ten high-power fields per sample were examined at 400× magnification using a 40× objective lens with a 10× eyepiece for quantitative analysis. Image analysis was performed using

**Fig. 6 | RNA sequencing and qRT-PCR validation in 3D culture, with integrated GSEA and Venn diagram analysis of 2D and 3D results. a** Heatmaps and volcano plots of DEGs (non-adjusted p < 0.05 and $\log_2$ fold change > 1 or < −1) between the RM$^+$ group and RM$^-$ group in 3D culture conditions (n = 4). Results of GO analysis are shown. The RM$^+$ group exhibited upregulation of genes potentially involved in cell proliferation and adhesion, such as *INHBA, LOX, LRRC15,* and CXCL12. No significant enrichment was observed in the Biological Process category. However, "extracellular space", "protein complex involved in cell–cell adhesion" and "extra-cellular region" were enriched under the Cellular Component category and "cyto-kine activity" under the Molecular Function category. **b** In the qRT-PCR analysis, no statistically significant differences were observed for *LOX, CXCL12,* or *CCND1*; however, *integrin β1* (p = 0.0277) and *CTGF* (p = 0.0296) were significantly upregulated in the RM$^+$ group (n = 4). Data are presented as mean ± SD, and error bars indicate the SD from three independent experiments. **c** GSEA based on the gene expression profiles in 3D culture revealed significant upregulation of the "androgen response" gene set in the RM$^+$ group (p < 0.0001) and "peroxisome" (p < 0.0001), "acid metabolism" (p < 0.0001), and "P53 pathway" (p = 0.0059) gene sets in the RM$^-$ group (n = 4). **d** Venn diagram of DEGs identified in 2D and 3D cultures. Only one gene was commonly upregulated across both models; no other common DEG was identified. qRT-PCR quantitative reverse transcription polymerase chain reaction; 3D three-dimensional, GSEA gene set enrichment analysis, 2D two-dimensional, DEG differentially expressed gene, RM respiratory motion, RM$^+$ with RM, RM$^-$ without RM, GO gene ontology, SD standard deviation, ns not significant. All error bars represent the standard deviation.

the ImageJ software (National Institutes of Health, https://imagej.nih.gov/ij/).

Images were first converted to greyscale for fluorescence intensity analysis, and background noise was subtracted. Ten regions of interest (ROIs) corresponding to intercellular boundaries were manually selected for each sample. Fluorescence intensity was measured, and comparisons between samples were performed using the corrected integrated density, accounting for background correction. The following formula was used:

Corrected integrated density = Raw integrated density − (Background mean × ROI area)

Furthermore, to normalise fluorescence intensity by cell number, nuclei within each ROI were visualised using DAPI staining and counted with the 'Analyze Particles' function in ImageJ. The corrected integrated density for each ROI was divided by the number of cells to calculate the fluorescence intensity per cell. The final normalised fluorescence intensity was expressed using the following formula:

Fluorescence intensity per cell = Corrected integrated density Number of cells in ROI

### Calculation of cell adhesion rate

Among all HE-stained sections prepared using the left lung, the pro-portion of the tissue area showing attached tumour cells was defined as the "cell adhesion rate". HE-stained slide images were imported into ImageJ software and converted to 8-bit greyscale images. The threshold was adjusted to visualise the attached cells. Based on the histogram displayed by ImageJ, the lower threshold (0%) was set for cell detection, and the upper threshold (100%) was set to include the entire tissue section.

The area detected at the 50% threshold, representing attached cells, was divided by the area detected at the 100% threshold, representing the entire tissue section. This ratio was used to calculate the cell adhesion rate as follows (Fig. 3c):

Cell adhesion rate (%) = (Area of detected cells at 50% threshold / Area of total section at 100% threshold) × 100

### RNA extraction and RNA sequencing

For 2D cultures, A549 cells were harvested from well plates using 0.05% trypsin–EDTA solution (Nacalai Tesque, Kyoto, Japan). For 3D cultures, a portion of the right lung was finely minced and immediately immersed in RNAlater Stabilisation Solution (Thermo Fisher Scientific, Waltham, MA, USA) following the manufacturer's instructions. The fixed lung samples were homogenised for at least 30 s using a BioMasher II homogeniser (Funakoshi, Tokyo, Japan).

Total RNA was extracted using the RNeasy Mini Kit (QIAGEN N.V., Venlo, The Netherlands) following the manufacturer's protocol. RNA concentration and purity were assessed using a NanoDrop One spectro-photometer (Thermo Fisher Scientific, Waltham, MA, USA). Samples were sent to Rhelixa (Tokyo, Japan) for RNA sequencing.

GSEA was performed using the RNA data obtained from 3D cultures. GSEA version 4.4.0 (https://www.gsea-msigdb.org/gsea/index.jsp) was used, with gene sets retrieved from the Molecular Signatures Database (MSigDB).

RNA was extracted from decellularised lungs (without cell seeding) using the same protocol to assess residual rat-derived RNA as a control (n = 1) (Supplementary Table 1).

### qRT-PCR

To validate genes potentially involved in the pathogenesis identified by RNA sequencing, qRT-PCR was performed. Total RNA (1 µg), fixed and pre-served using the aforementioned method, was reverse transcribed into complementary DNA (cDNA) using ReverTra Ace qPCR RT Master Mix (TOYOBO CO., LTD., Osaka, Japan) in accordance with the manufacturer's instructions. The reverse transcription reaction was carried out using a Takara PCR Thermal Cycler Dice Gradient (Takara Bio Inc., Shiga, Japan) with the following temperature protocol: 37 °C for 15 min, 50 °C for 5 min, and 98 °C for 5 min.

qRT-PCR was conducted using 96-well plates (BIO-BIK Ina Optika CO., Ltd., Osaka, Japan), with each well containing 2X Brilliant III Ultra-Fast SYBR Green QPCR Master Mix (Agilent, CA, USA), target-specific primers, and 1 µL of cDNA, adjusted with RNase-free water to a final volume of 24.5 µL per well. The sequences of the primers used for qRT-PCR are listed in Supplementary Table 2. Amplification was performed on the AriaMx Real-Time PCR System (Agilent, CA, USA), and data were analysed using GENETYX software (Nihon Server Corporation, Tokyo, Japan). The qRT-PCR cycling conditions were as follows: 95 °C for 15 min; 40 cycles of 94 °C for 10 s, 55 °C for 30 s, and 72 °C for 30 s; followed by 72 °C for 10 min.

GAPDH was used as a housekeeping gene to calculate ΔCt values. Relative expression was determined using the ΔΔCt method, and fold changes were calculated as $2^{-\Delta\Delta Ct}$. Note that while the mean ΔΔCt of the control group is 0 by definition, the mean fold change ($2^{-\Delta\Delta Ct}$) may differ from 1 due to the non-linearity of the transformation.

### Statistics and reproducibility

Fisher's exact test was used for categorical variables. Welch's t-test was used to compare continuous variables between the two groups. A *p*-value of less than 0.05 was considered statistically significant. A $-\log_{10}$(*p*-value) greater than 1.30 was considered statistically significant. All statistical analyses were performed using JMP Pro software (version 16.2.0; JMP Statistical Dis-covery LLC, Cary, NC, USA).

### Institutional review board statement

All animal experiments were approved by the Animal Experimentation Committee of the University of Toyama (approval number: A2023UH-01).

### Reporting summary

Further information on research design is available in the Nature Portfolio Reporting Summary linked to this article.

### Data availability

The RNA-seq data generated in this study have been deposited in the Gene Expression Omnibus (GEO) under accession numbers GSE309015 and GSE309016, and are publicly available. The processed data, including lists of differentially expressed genes, as well as additional supporting datasets, have been deposited in Figshare (https://doi.org/10.6084/m9.figshare.30203953)

**Article**

and are publicly available. All other data are available from the corresponding author upon reasonable request.

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

## Acknowledgements

The authors gratefully acknowledge the technical support provided by Sanae Hirota (University of Toyama) and the administrative assistance provided by Ruriko Ishisaka (University of Toyama). This work was supported by the Grant-in-Aid for Research Activity Start-up (Project title: Analysis of mechanical stress in ex vivo lung cancer models; Grant number: 23K19530; FY2023–2024) and by a donation from Hayashida Finance LLC.

## Author contributions

N.Ki. conceived and designed the study, performed experiments, collected and analyzed data, and wrote the manuscript. M.I. performed experiments, collected data, and supervised experimental procedures. S.I.M. and K.S. performed experiments and supervised experimental procedures. S.W., K.K., M.H., V.D.N., and R.Y. performed experiments and collected data. N. Ko. and T.O. performed experiments and collected data. N.K. contributed to the design, fabrication, and improvement of experimental devices and assisted with data collection. H.H. and K.H. prepared pathological slides; K.H. also supervised the study. Y.M. performed experiments, collected data, supervised experimental procedures, and contributed to study supervision. M.S.B.R. supervised the study and contributed to data analysis and interpretation. T.T. conceived and designed the study, supervised the study, and revised the manuscript. All authors reviewed and approved the final manuscript.

## Competing interests

The authors declare no competing interests.
