## [Transparent Peer Review file · Communications Biology]

Lung adenocarcinoma cells respond differently to mechanical stress in 3D versus 2D environments

Corresponding Author: Dr Naoya Kitamura

This manuscript has been previously submitted at another journal. This document only contains information relating to versions considered at Communications Biology.

Version 0:

Reviewer comments:

Reviewer #1

(Remarks to the Author)

The submitted work described an in vitro model of lung adenocarcinoma by using decellularized rat lungs matrix and the A549 cell line. The novelty of the paper relies on the use of a pressure stimulation unit to mimic the pressure derived from respiratory motion. Although similar systems have been already studied, the combination of 3D and dynamic culture conditions has the potential to improve the knowledge of a complex tissue like the lungs. The authors successfully proved that the designed system increased the seeding of the cells on the scaffold, also modulated protein and gene expression compared to 2D or a non-stimulated system. I think that after revision to clarify some results, the work should be taken into consideration for publication.

My major comments are listed below:

- 1) All experiments were tested at 4 days. How is this timepoints chosen?
- 2) In the line 130, the overnight preconditioning is referred to A549? In that case, the authors should describe it in the method section, because could be confused with the decellularization of the rat lungs.
- 3) In Figure 1, the scale bar is not indicated. In the Ki67 and in the Cleaved-caspase 3 staining, a legenda could help the reader in understanding the described results.
- 4) Regarding Figure 1g, in the text was claimed that no YAP translocation was detected, but, from the proposed pictures, this is not so clear. Could the authors add a quantification of nuclear and Yap staining co-localization?
- 5) In figure 4e, the differences in integrin beta1 expression are not-entirely convincing. While the authors quantified the signal, the presented images do not fully support the results shown in the graphs; a different picture might better illustrate these differences. A protein quantification by western blot analysis would significantly strengthen the demonstration of this modulated expression.
- 6) Overall a deeper discussion on the results of gene expression analysis is needed.

Reviewer #2

(Remarks to the Author)

This manuscript investigates how lung adenocarcinoma cells respond to mechanical stress in different dimensional culture environments, focusing on the physiological relevance of a three-dimensional (3D) ex vivo lung model. By seeding human A549 lung cancer cells into decellularised rat lungs and subjecting them to simulated respiratory motion in a custom-designed bioreactor, the authors demonstrate that mechanical forces, such as respiratory stretch and vascular shear stress, significantly influence cancer cell behavior. Comparative analyses with conventional two-dimensional (2D) cultures reveal that mechanical stress in 2D suppresses proliferation and promotes apoptosis, whereas in the 3D model, respiratory motion enhances cell adhesion, proliferation, and nuclear translocation of β -catenin and YAP. Transcriptomic profiling further underscores the context-dependent nature of mechanotransduction, with divergent gene expression profiles observed under identical mechanical stimuli across dimensional settings. The study claims to establish a more physiologically relevant

cancer model that may bridge the gap between in vitro studies and in vivo tumor biology by incorporating native ECM architecture and biomechanical cues absent in traditional systems.

1. Figure 1c is missing a scale bar. Please include it for proper spatial interpretation.
2. In Figure 1f, cleaved caspase-3 staining appears present in RM⁻ as well, just at lower intensity. The apparent difference may be mostly contrast-related, not a true absence of signal.
3. Figures 1e–f and 4a–b show n = 3–4, but the scatter plots have more dots than expected. If multiple fields per sample were used, that should be clarified with a note.
4. Figures 2b–c don't quantify the actual mechanical forces. How close are the applied stretch and perfusion to physiological shear and respiratory motion? Some estimate or reference is needed.
5. Figure 3 is interesting, but it would be more convincing with IF showing actual cell–ECM interactions—e.g., integrin, fibronectin, or focal adhesion staining.
6. Figures 4c–d have too low resolution to clearly confirm nuclear localization. DAPI overlap is hard to assess. Higher-res (e.g., confocal) images are needed to support nuclear translocation claims.
7. Figure 4f uses field-wide intensity values, not per-cell. That's vulnerable to bias from differences in cell density. Normalizing per cell would improve the analysis.
8. Figure 5 has relatively few DEGs, and only unadjusted p < 0.05 was used—no FDR. Most GO terms weren't significant. These data should be interpreted with care.
9. No validation (e.g., qPCR or WB) of RNA-seq data is shown. Without this, the transcriptomic results remain preliminary. Follow-up experiments are needed to support these claims.
10. Minor comments:
 - The abbreviation "RM" for respiratory motion is not defined at first mention in the Abstract.
 - Inconsistent use of "respiratory motion (RM)" and "RM⁺/RM⁻" throughout the text; some figures and legends do not clarify which condition they refer to.
 - In Figure 1 caption, "Ki-67 staining" should be written as "Immunostaining for Ki-67" for clarity and consistency.
 - Figure resolution is insufficient in several panels, particularly Figure 4a and 4c; nuclear localization of YAP and β -catenin is hard to discern.
 - Some scale bars are missing or inconsistently labeled (e.g., Figure 3b and 4c).
 - In Figure 5, gene names in volcano plots are hard to read due to excessive label overlap.
 - The term "bioreactor" is used variably without specification—pressure chamber vs perfusion system should be clearly distinguished.
 - In the Methods, "recellularised" and "re-cellularized" are both used; spelling should be standardized.
 - "the lung undergoes periodic mechanical stresses" should be "lungs undergo" for grammatical agreement.
 - "the generated model" is vague—should specify "the 3D decellularised lung cancer model".
 - The legend of Supplementary Figure 1 does not specify magnification or scale for the HE images.
 - "Decellularised" is used throughout the text, but the American spelling "decellularized" appears in Figure legends.
 - In Supplementary Table 1, "Decellularised lung" has values with only one decimal place, while others have three—numerical precision should be consistent.
 - Abbreviation "ECM" is overused without contextual reminder, especially in the Discussion section.
 - In the References, several journal names are inconsistently italicized or abbreviated (e.g., Sci. Rep. vs Scientific Reports).
 - The legend for Figure 2 does not define abbreviations like BPU and PSU upon first appearance.
 - The phrase "biologically relevant" is used repeatedly without clarification—consider substituting with more precise terminology.

Version 1:

Reviewer comments:

Reviewer #1

(Remarks to the Author)

The authors have addressed all my concerns, implementing some of the analysis that initially appeared less robust. I thank the authors for their work and after the revision, I believe that the paper should be accepted for publication.

Reviewer #2

(Remarks to the Author)

All of the reviewers' concerns have been thoroughly and thoughtfully addressed in the revised manuscript. The authors have made substantial improvements in clarity, experimental validation, and data presentation. I find the responses satisfactory, and the manuscript is now suitable for publication. I would like to thank the authors for their careful revisions and commend their effort in strengthening the work.

Response to the Reviewers' Comments

Reviewer #1

We sincerely thank the Editor and the Reviewers for their thoughtful and constructive comments. We have carefully revised our manuscript in response to all suggestions. Below, we provide a detailed point-by-point response, with corresponding changes incorporated into the revised manuscript.

1) All experiments were tested at 4 days. How is this timepoints chosen?

Response:

We thank the reviewer for this important question. In this study, we selected day 4 as the evaluation time point based on the following rationale: it represents a compromise between the shorter time scale of transcriptional responses (24–48 h) and the longer time scale of cell adhesion and tissue organization (several days to one week).

1. Time scale of gene expression (short-term: 24–48 h).

Previous reports in A549 cells have shown that mechanical stress induces a transient, acute response such as c-fos mRNA, which peaks within 30 minutes¹. In addition, inflammatory-related genes were significantly upregulated after approximately 26 h of mechanical stimulation². These studies indicate that changes in gene expression can be detected within 1–2 days.

2. Time scale of adhesion and tissue organization (mid-term: several days to one week).

In a rat decellularized lung recellularization model, cell distribution and viability were optimal around day 7–8, whereas apoptosis progression and decreased viability were observed by day 16³. Similarly, in a mixed recellularization model including AEC2s, fibroblasts, and endothelial cells, the most prominent cell adhesion was observed at day 6⁴, suggesting that adhesion maturation requires several days.

Taken together, day 4 was considered an appropriate midpoint because:

- it is a time when acute transcriptional responses (within 24–48 h) have already emerged and begun to stabilize;
- adhesion and morphological reorganization are underway and thus can be evaluated; and
- it precedes the adverse effects of longer-term culture (>1 week) on cell viability.

Accordingly, we created a new subsection entitled “*Rationale for the time point for evaluation*” in the Methods section to clarify this choice.

References

1. Ying, B. et al. Mechanical strain-induced c-fos expression in pulmonary epithelial cell line A549. *Biochem. Biophys. Res. Commun.* **347**, 369–372 (2006). [10.1016/j.bbrc.2006.06.105](https://doi.org/10.1016/j.bbrc.2006.06.105), PubMed: [16815297](https://pubmed.ncbi.nlm.nih.gov/16815297/).
2. Charles, P.-E. et al. Mild-stretch mechanical ventilation upregulates toll-like receptor 2 and sensitizes the lung to bacterial lipopeptide. *Crit. Care* **15**, R181 (2011). [10.1186/cc10330](https://doi.org/10.1186/cc10330), PubMed: [21794115](https://pubmed.ncbi.nlm.nih.gov/21794115/).
3. Doi, R. et al. Transplantation of bioengineered rat lungs recellularized with endothelial and adipose-derived stromal cells. *Sci. Rep.* **7**, 8447 (2017). [10.1038/s41598-017-09115-2](https://doi.org/10.1038/s41598-017-09115-2), PubMed: [28814761](https://pubmed.ncbi.nlm.nih.gov/28814761/).
4. Leiby, K. L. et al. Rational engineering of lung alveolar epithelium. *npj Regen. Med.* **8**, 22 (2023). [10.1038/s41536-023-00295-2](https://doi.org/10.1038/s41536-023-00295-2), PubMed: [37117221](https://pubmed.ncbi.nlm.nih.gov/37117221/).

2) In the line 130, the overnight preconditioning is referred to A549? In that case, the authors should describe it in the method section, because could be confused with the decellularization of the rat lungs.

Response:

We appreciate the reviewer's comment and apologise for the ambiguity in our original description. The term "preconditioning" in this context refers to intravascular perfusion aimed at promoting peripheral vascular expansion of the decellularised lungs, not to any procedure involving A549 cells. Immediately after decellularisation, the vasculature tends to remain constricted, leading to an abnormal rise in intravascular pressure even with a small volume of perfusion. To address this, we performed overnight perfusion of culture medium within an incubator, which served as a preconditioning step to reduce vascular resistance and ensure adequate perfusion and nutrient delivery.

Although this procedure was briefly mentioned in the Methods section (lines 385–389), we recognise that its purpose was insufficiently explained. To avoid confusion, we have now revised both the Results and Methods sections for clarity, as follows:

Results ("3D cell culture using pressure chamber and perfusion system"):

- Original: *"After overnight preconditioning, A549 cells were used for recellularisation."*
- Revised: *"Following overnight preconditioning to promote peripheral vascular expansion of the decellularised lungs (i.e., continuous intravascular perfusion with culture medium at a constant flow rate), recellularisation was performed using A549 cells."*

Methods ("Seeding and perfusion culture of lung cancer cells"):

- Original: *"Two decellularised lungs and corresponding chambers were prepared to compare conditions with and without respiratory motion. On the evening of Day 0, preconditioning of the decellularised lungs was initiated to promote peripheral vascular expansion. Specifically, each lung was placed in a chamber filled*

with 200 mL of DMEM and connected to vascular perfusion at a flow rate of 14 mL/min in an incubator at 37 °C with 5% CO₂. During this period, neither chamber was subjected to respiratory motion.”

• Revised: *“Two decellularised lungs and corresponding chambers were prepared to compare conditions with and without RM. On the evening of day 0, vascular preconditioning of the decellularised lungs was initiated to promote peripheral vascular expansion prior to cell seeding. Specifically, each lung was placed in a chamber filled with 200 mL of DMEM and connected to vascular perfusion at a flow rate of 14 mL/min in an incubator at 37 °C with 5% CO₂. This procedure was intended to prevent inadequate perfusion caused by vasoconstriction following decellularisation and was unrelated to either the decellularisation process or any cellular intervention. During this period, neither chamber was subjected to RM.”*

Figure legends, Figure 2 (c):

• Original: *“intravascular perfusion (14 mL/min) was initiated for preconditioning.”*

• Revised: *“intravascular perfusion (14 mL/min) was initiated to promote peripheral vascular expansion in the lung.”*

We believe these revisions make the rationale and purpose of vascular preconditioning clearer, and eliminate potential confusion with the decellularisation process or with A549 cell preparation.

3) In Figure 1, the scale bar is not indicated. In the Ki67 and in the Cleaved-caspase 3 staining, a legend could help the reader in understanding the described results.

Response:

We thank the reviewer for pointing this out. We have added appropriate scale bars to Figure 1c as well as to other images where they were missing (e.g., Fig. 3a and 3b). In addition, we have inserted legends (labels) within the figures for (e) Ki-67 and (f) Cleaved caspase-3 in Figure 1, as well as for (a) Ki-67 and (b) Cleaved caspase-3 in Figure 4, to facilitate interpretation of the results.

4) Regarding Figure 1g, in the text was claimed that no YAP translocation was detected, but, from the proposed pictures, this is not so clear. Could the authors add a quantification of nuclear and Yap staining co-localization?

Response:

We thank the reviewer for this valuable comment and apologise for presenting images that were unclear. Upon re-examining the contrast, we confirmed that nuclear translocation of YAP could be observed in both the RM⁺ and RM⁻ groups. To address this concern, we quantified the proportion of cells with nuclear localisation by counting 10 randomly selected fields per sample, in line with the approach used for other immunostaining analyses.

We have replaced the images with higher-magnification views to make nuclear staining more evident and added a quantitative graph of the nuclear co-localisation (Fig. 1g). This analysis showed no significant difference between the two groups ($27.27 \pm 8.77\%$ vs $27.55 \pm 9.65\%$, $p = 0.905$, $n = 3$).

5) In figure 4e, the differences in integrin beta1 expression are not-entirely convincing. While the authors quantified the signal, the presented images do not fully support the results shown in the graphs; a different picture might better illustrate these differences. A protein quantification by western blot analysis would significantly strengthen the demonstration of this modulated expression.

Response:

We thank the reviewer for this insightful comment and apologise for the insufficient images originally presented. For Figure 4e, we have replaced the images with more appropriate ones that better illustrate the differences between the two groups. We also performed supplementary qRT-PCR experiments, which yielded results supporting the observed difference in integrin β 1 expression. These results have been incorporated into the revised manuscript, providing additional evidence to reinforce the difference in integrin β 1 expression (Fig. 6b). While we acknowledge that we could not fully comply with the reviewer's suggestion, we believe that the supplementary data have strengthened the overall conclusions of our study.

6) Overall a deeper discussion on the results of gene expression analysis is needed.

Response:

We thank the reviewer for this constructive suggestion. In addition to the RNA sequencing results, we incorporated qRT-PCR data into the revised manuscript. Under 2D conditions, the RM⁺ group showed significant upregulation of the tumour-suppressive genes *CDKN1A* and *NR4A3*, whereas under 3D conditions, *integrin β 1* and *CTGF* were upregulated. These results suggest that dimensionality influences how cells respond to mechanical stress: in 2D, responses are directed towards cell-cycle inhibition and stress regulation, while in 3D, ECM-dependent mechanosensing and activation of the YAP/TAZ pathway predominate. The concordant upregulation of *integrin β 1* and *CTGF* in 3D is consistent with integrin-mediated force transmission leading to YAP/TAZ activation and CTGF induction, supporting a framework of adaptation through ECM remodelling.

We also noted that not all RNA-seq and qRT-PCR results were fully concordant, likely due to differences in thresholds, sample size, or post-transcriptional regulation; this limitation is now acknowledged. Finally, we expanded the Discussion to highlight potential future directions, including integrin β 1 or YAP inhibition and traction force measurements, as well as the clinical implications of mechanical cues in shaping tumour microenvironments.

Reviewer #2

We sincerely thank Reviewer #2 for the thoughtful and constructive comments. The suggestions greatly helped us to clarify our methodology and to strengthen the interpretation of our findings. We have carefully addressed all points raised, and we believe the manuscript has been substantially improved as a result.

1. Figure 1c is missing a scale bar. Please include it for proper spatial interpretation.

Response:

We thank the reviewer for this comment. We have added an appropriate scale bar to Figure 1c. In addition, scale bars have been inserted into other images where they were previously missing (e.g., Fig. 3a and 3b).

2. In Figure 1f, cleaved caspase-3 staining appears present in RM⁻ as well, just at lower intensity.

Response:

The apparent difference may be mostly contrast-related, not a true absence of signal. We thank the reviewer for this important comment. It has been reported that A549 cells exhibit a basal level of apoptosis even under standard culture conditions ($2.65 \pm 0.31\%$) (Ho, Y.F., Karsani, S.A., Yong, W.K. & Abd Malek, S.N. Induction of apoptosis and cell cycle blockade by helichrysetin in A549 human lung adenocarcinoma cells. *Evid. Based Complement. Alternat. Med.* 2013, 857257 (2013)). Therefore, it is reasonable that a certain proportion of apoptotic cells is also observed in the RM⁻ group of our study. After re-examining the images to avoid contrast-related artefacts, we confirmed the presence of cleaved caspase-3–positive cells in the RM⁻ group as well.

We have re-performed the statistical analysis and revised the graphs, images (Fig. 1f), and corresponding text accordingly. Importantly, the conclusion remains unchanged: the difference between the two groups remains statistically significant ($25.07 \pm 11.59\%$ vs $8.07 \pm 4.27\%$; $p < 0.0001$, $n = 3$).

3. Figures 1e–f and 4a–b show $n = 3$ –4, but the scatter plots have more dots than expected. If multiple fields per sample were used, that should be clarified with a note.

Response:

We apologise for the lack of clarity in our original description. To address this, we have added the following notes to the figure legends:

Figure 1:

“Note: For (e–g), each biological sample ($n = 3$) was measured in 10 randomly selected microscopic fields (regions of interest), and all data points from these fields are shown in the scatter plots.”

Figure 4:

“Note: For panels (a–f), each biological sample ($n = 4$) was measured in 10 randomly selected microscopic fields (regions of interest), and all data points from these fields are displayed in the scatter plots.”

We believe these clarifications will help the reader correctly interpret the scatter plots in Figures 1 and 4.

4. Figures 2b–c don't quantify the actual mechanical forces. How close are the applied stretch and perfusion to physiological shear and respiratory motion? Some estimate or reference is needed.

Response:

We thank the reviewer for raising this important point. The mean pulmonary arterial pressure in healthy individuals has been reported to be approximately 14.0 ± 3.3 mmHg, with little variation across sex or ethnicity¹. According to current diagnostic criteria, pulmonary hypertension is defined as a mean pulmonary arterial pressure exceeding 20 mmHg². In our model, the intravascular pressures measured on days 3–4 were 10.47 ± 6.40 mmHg (RM⁺) and 12.93 ± 4.44 mmHg (RM⁻), indicating that the perfusion conditions reproduced physiological pulmonary arterial pressures and corresponding vascular shear stress in vivo.

Regarding respiratory motion, intrapleural pressure is typically around -5 cmH₂O (≈ -3.7 mmHg) and reaches approximately -7.5 cmH₂O (≈ -5.5 mmHg) during inspiration, as described in standard physiology textbooks³. Direct measurements in healthy individuals have also reported end-expiratory pressures of approximately -2.5 to -3 mmHg⁴. Based on these reports, we set the chamber pressure to fluctuate between 0 and -5 mmHg. As shown in Figures 1b and 2c, this cyclic negative pressure closely approximates the physiological intrathoracic pressure changes observed during spontaneous breathing.

Taken together, the stretch and perfusion conditions applied in our model fall within the physiological ranges of pulmonary vascular and intrathoracic pressures. We have added the relevant references and clarified these points in the Methods section (*Seeding and perfusion culture of lung cancer cells*). On the other hand, to avoid redundancy, the corresponding description in the Discussion (*Bioreactor mimicking a physiological environment*) was kept concise as follows: *In our study, a bioreactor equipped with a pressure chamber and perfusion system successfully provided an environment in which both vascular and intrathoracic pressures remained within the physiological ranges reported in humans.*

References

1. Kovacs, G., Berghold, A., Scheidl, S. & Olschewski, H. Pulmonary arterial pressure during rest and exercise in healthy subjects: a systematic review. *Eur. Respir. J.* **34**, 888–894 (2009). [10.1183/09031936.00145608](https://doi.org/10.1183/09031936.00145608), PubMed: [19324955](https://pubmed.ncbi.nlm.nih.gov/19324955/).
2. Simonneau, G. et al. Haemodynamic definitions and updated clinical classification of pulmonary hypertension. *Eur. Respir. J.* **53**, 1801913 (2019). [10.1183/13993003.01913-2018](https://doi.org/10.1183/13993003.01913-2018), PubMed: [30545968](https://pubmed.ncbi.nlm.nih.gov/30545968/).
3. Hall, J. E. *Guyton and Hall Textbook of Medical Physiology*. 13th ed 498 (Elsevier, 2016).
4. Daly, W. J. & Bondurant, S. Direct measurement of respiratory pleural pressure changes in normal man. *J. Appl. Physiol.* (1985) **18**, 513–518 (1963). [10.1152/jappl.1963.18.3.513](https://doi.org/10.1152/jappl.1963.18.3.513), PubMed: [31094492](https://pubmed.ncbi.nlm.nih.gov/31094492/).

5. Figure 3 is interesting, but it would be more convincing with IF showing actual cell–ECM interactions—e.g., integrin, fibronectin, or focal adhesion staining.

Response:

We thank the reviewer for this valuable suggestion to strengthen the evidence for cell–ECM interactions. In response, we have performed additional dual immunofluorescence staining of integrin $\beta 1$ with collagen I (+ DAPI) and integrin $\beta 1$ with fibronectin (+ DAPI), which are now included in the revised manuscript (Fig. 4g, h).

Consistent with the increased expression of integrin $\beta 1$, in the RM^+ group we could glimpse a tendency toward co-localisation of integrin $\beta 1$ with collagen I and with fibronectin. Together with the immunostaining and qRT-PCR findings showing upregulation of integrin $\beta 1$ in the RM^+ group, these results provide stronger evidence of enhanced cell–ECM interactions (alveolar wall adhesion) and support the conclusion that cellular adhesion was promoted under RM^+ conditions.

The Results section (*3D cell culture using a pressure chamber and perfusion system*) has been revised accordingly, with the following description:

“Dual immunofluorescence staining of integrin $\beta 1$ and collagen I, and DAPI counterstaining, showed evidence of cellular adhesion to the ECM mediated by integrin $\beta 1$ and collagen I in the RM^+ group, concomitant with increased integrin $\beta 1$ expression (Fig. 4g). Similarly, dual immunofluorescence staining of integrin $\beta 1$ and fibronectin, and DAPI counterstaining, revealed integrin $\beta 1$ and fibronectin-mediated cellular adhesion to the ECM in the RM^+ group (Fig. 4h).”

6. Figures 4c–d have too low resolution to clearly confirm nuclear localization. DAPI overlap is hard to assess. Higher-res (e.g., confocal) images are needed to support nuclear translocation claims.

Response:

We apologise for the unclear images that hindered interpretation. We have replaced Figures 4c and 4d with alternative images at higher magnification and improved resolution, making the nuclear localisation and DAPI overlap more clearly discernible.

7. Figure 4f uses field-wide intensity values, not per-cell. That's vulnerable to bias from differences in cell density. Normalizing per cell would improve the analysis.

Response:

We thank the reviewer for this insightful comment. As suggested, we re-analysed the data after normalising fluorescence intensity on a per-cell basis and replaced Figure 4f with an updated version that more clearly illustrates the difference between groups.

In the Methods section (Image acquisition and intensity normalisation for quantification), we have added the following description:

“Furthermore, to normalise fluorescence intensity by cell number, nuclei within each ROI were visualised using DAPI staining and counted with the ‘Analyze Particles’ function in ImageJ. The corrected integrated density for each ROI was divided by the number of cells to calculate the fluorescence intensity per cell. The final normalised fluorescence intensity was expressed using the following formula:

Fluorescence intensity per cell = Corrected integrated density / Number of cells in ROI”

After normalization, fluorescence intensity remained significantly higher in the RM⁺ group compared with the RM⁻ group (10,167 ± 3,277 vs 8,530 ± 2,422 per cell, p = 0.0132, n = 4) (Fig. 4f). Importantly, this re-analysis did not alter the interpretation of our results, which continue to support enhanced fluorescence intensity under RM⁺ conditions.

8. Figure 5 has relatively few DEGs, and only unadjusted $p < 0.05$ was used—no FDR. Most GO terms weren't significant. These data should be interpreted with care.

Response:

We thank the reviewer for this important comment. As noted, the DEGs identified in our study were based on $p < 0.05$ without multiple testing correction. We agree that these results should be interpreted with caution and regarded as exploratory in nature. Accordingly, we have revised the Discussion (*Limitations*) to adopt a more conservative tone, including the following statement:

“The DEGs identified in this study were based on $p < 0.05$ without multiple testing correction and should therefore be interpreted cautiously, acknowledging their exploratory nature.”

9. No validation (e.g., qPCR or WB) of RNA-seq data is shown. Without this, the transcriptomic results remain preliminary. Follow-up experiments are needed to support these claims.

Response:

We thank the reviewer for this valuable comment to strengthen the manuscript. To validate the RNA-seq results, we performed additional qRT-PCR experiments.

In the 2D environment with rigid substrate, qRT-PCR confirmed significant upregulation of the tumour-suppressive genes *CDKN1A* and *NR4A3* in the RM⁺ group. In contrast, under 3D conditions with alveolar-derived scaffolds, the RM⁺ group showed significant upregulation of *integrin β1* and *CTGF*, the latter a well-established YAP target gene induced by nuclear translocation of YAP.

These validation data are consistent with the trends indicated by RNA-seq, thereby reinforcing the interpretation that RM promotes the expression of tumour-suppressive genes in 2D environments, while enhancing the expression of adhesion-related genes in 3D environments. This suggests that, while the stress response and cell-cycle inhibition predominate under 2D conditions, ECM-dependent mechanosensing and YAP/TAZ activation become more prominent in 3D conditions.

We have incorporated these additional results into the Results section (*RNA sequencing and quantitative reverse transcription polymerase chain reaction (qRT-PCR)*) and expanded the Discussion (particularly in the subsections *Comparison between 2D and 3D models* and *Effects of RM in the 3D model*) to provide a more detailed interpretation.

10. Minor comments:

- The abbreviation “RM” for respiratory motion is not defined at first mention in the Abstract.

Response:

We thank the reviewer for pointing this out and apologise for the lack of clarity. In the Abstract, the term *respiratory motion* had been written out in full on each occasion, without using the abbreviation “RM.” To ensure consistency with the main text, we have now defined it at first mention as “*respiratory motion (RM)*” and used the abbreviation “RM” thereafter.

- Inconsistent use of “respiratory motion (RM)” and “RM⁺/RM⁻” throughout the text; some figures and legends do not clarify which condition they refer to.

Response:

We thank the reviewer for this helpful comment. “RM⁺” indicates the presence of respiratory motion and “RM⁻” indicates the absence of respiratory motion; however, as noted, clarification is necessary for readers. In the main text, we now define “respiratory motion (RM)” at the first mention and subsequently use the abbreviation “RM.” Since the terms “RM⁺” and “RM⁻” appear in all figures, we have added the following explanatory note to every figure legend: “*RM, respiratory motion; RM⁺, with RM; RM⁻, without RM.*” In addition, in Supplementary Table 1, where the terms “RM” and “non-RM” were inconsistently used, we have added a definition and unified the terminology to “RM⁺” and “RM⁻.”

- In Figure 1 caption, “Ki-67 staining” should be written as “Immunostaining for Ki-67” for clarity and consistency.

Response:

We thank the reviewer for this suggestion. In accordance with the recommendation, we have revised Figure 1e to read “*Immunostaining for Ki-67.*” For consistency, we have also revised Figure 1f to “*Immunostaining for cleaved caspase-3.*”

- Figure resolution is insufficient in several panels, particularly Figure 4a and 4c; nuclear localization of YAP and β -catenin is hard to discern.

Response:

We apologise that the images were not sufficiently clear. To enable the nuclear localisation of YAP and β -catenin to be more clearly discerned, we have replaced Figures 4c and 4d, as well as Figures 1g and 1h, with higher-magnification images.

- Some scale bars are missing or inconsistently labeled (e.g., Figure 3b and 4c).

Response:

We thank the reviewer for pointing this out. We have added consistent scale bars to all relevant panels. Specifically, a 5 mm scale bar was added to Figures 3a and 3b to match the adjacent pathological images. Scale bars were also added to Figures 1g and 1h (25 μ m) and Figures 4c and 4d (50 μ m), ensuring consistent labelling across panels. In addition, missing scale bars were inserted in Figures 1c and 2d.

- In Figure 5, gene names in volcano plots are hard to read due to excessive label overlap.

Response:

We apologise for the confusing presentation, as some gene names upregulated in 3D (right side) were incorrectly placed on the downregulated side (left side). We have revised Figure 5 (Figures 5 and 6 in the revised version) by adjusting the font size and repositioning the gene labels to avoid overlap, thereby

improving readability.

- The term “bioreactor” is used variably without specification—pressure chamber vs perfusion system should be clearly distinguished.

Response:

We thank the reviewer for this helpful comment. To improve clarity, we have revised the terminology throughout the manuscript. In the Abstract, Introduction, Results, and Methods, the term “*bioreactor*” has either been replaced with “*pressure chamber and perfusion system*” or, where retained, has been more precisely defined. Where appropriate, we also added clarifications when using the term “*chamber*.”

Examples of revisions include the following:

Results

- Original: “*2D Cell Culture Using a Bioreactor*”
- Revised: “*2D cell culture using a pressure chamber*”

- Original: “*a pressure stimulation unit (PSU; TOKAIHIT, Shizuoka, Japan)*”
- Revised: “*a pressure stimulation unit (PSU; TOKAIHIT, Shizuoka, Japan) served as a pressure chamber;*”

- Original: “*3D Cell Culture Using a Bioreactor*”
- Revised: “*3D cell culture using a pressure chamber and perfusion system*”

• Original: “Only the RM^+ group was subjected to pressure changes inside the sealed chamber via the PSU, establishing a bioreactor circuit similar to that used in 2D culture (Fig. 2b).”

• Revised: “Only the RM^+ group was subjected to pressure changes inside the sealed chamber via the PSU, undergoing pressure regulation similar to that used in the 2D culture system (Fig. 2b).”

Methods

• Original: “Bioreactor Setup for the 3D Culture”

• Revised: “Pressure chamber and perfusion system setup for the 3D culture”

• Original: “Bioreactor Setup for the 2D Culture”

• Revised: “Pressure chamber setup for the 2D culture”

• Original: “2D culture chamber”

• Revised: “2D culture pressure chamber”

These revisions ensure that “*bioreactor*” is used consistently and unambiguously throughout the manuscript.

In the Methods, “recellularised” and “re-cellularized” are both used; spelling should be standardized.

Response:

We apologise for the inconsistency in terminology. We have carefully reviewed the manuscript and standardised the spelling throughout, unifying it to the British English form “*recellularised*.” In addition, we checked the manuscript for other spelling inconsistencies (e.g., *organisation/organization*) and revised them to

maintain consistency with British English usage.

- “the lung undergoes periodic mechanical stresses” should be “lungs undergo” for grammatical agreement.

Response:

We thank the reviewer for this suggestion. As recommended, we have revised the sentence in the Introduction as follows:

• Original: “*Notably, the lung undergoes periodic mechanical stresses due to respiration, exhibiting anisotropic and heterogeneous large deformations.*”

• Revised: “*Notably, lungs undergo periodic mechanical stresses due to respiration, exhibiting anisotropic and heterogeneous large deformations.*”

- “the generated model” is vague—should specify “the 3D decellularised lung cancer model”.

Response:

We thank the reviewer for this helpful comment. In the Introduction, we have revised the phrasing for clarity as follows:

• Original: “*Using the generated model,*”

• Revised: “*Using the 3D decellularised lung cancer model,*”

- The legend of Supplementary Figure 1 does not specify magnification or scale for the HE images.

Response:

We thank the reviewer for pointing this out. We have added scale bars to Supplementary Figures 1a and 1b. For Supplementary Figures 1c and 1d, we have included the magnification both alongside the images and in the figure legend.

- “Decellularised” is used throughout the text, but the American spelling “decellularized” appears in Figure legends.

Response:

We thank the reviewer for pointing this out. We carefully re-checked both the main text and figure legends and have standardised the spelling throughout, unifying it to “*decellularised*.”

- In Supplementary Table 1, “Decellularised lung” has values with only one decimal place, while others have three—numerical precision should be consistent.

Response:

We thank the reviewer for highlighting this inconsistency. In Supplementary Table 1, RNA values for the decellularised lung were originally shown with two decimal places, while other values were presented with one decimal place, and the sample “non-RM_2” had no decimal places displayed. We have corrected this by revising all numerical values to one decimal place for consistency.

- Abbreviation “ECM” is overused without contextual reminder, especially in the Discussion section.

Response:

We thank the reviewer for this valuable suggestion to improve the clarity of our manuscript. As noted, the abbreviation “ECM” was overused without sufficient contextual explanation, particularly in the Discussion, and the distinction between alveolar and vascular ECM should have been made clearer. To address this, we

revised the text by adding explanatory details, rephrasing expressions, or removing the term “ECM” where appropriate. The specific revisions are shown below:

Introduction

- Original: *“This process retains the vascular network, yielding a complex ECM serving as a 3D scaffold, which has been refined and applied in tissue engineering and regenerative medicine.”*
- Revised: *“This process preserves both the alveolar and vascular scaffolds, yielding a complex ECM that functions as a 3D support structure, which has been applied in tissue engineering and regenerative medicine.”*

Discussion

- Original: *“Our findings emphasise the importance of incorporating respiratory motion and an ECM that supports mechanosensing to investigate lung cancer biology more accurately.”*
- Revised: *“Our findings emphasise the importance of incorporating RM in a 3D ECM that supports mechanosensing to investigate lung cancer biology more accurately.”*

Comparison between 2D and 3D models

- Original: *“in a 2D environment lacking ECM,”*
- Revised: *“in a 2D environment, where only a rigid substrate is provided and the native softness of the tissue is not recapitulated,”*
- Original: *“These results suggest that in a 3D model with ECM support, negative pressure does not act as a stressor;”*

- Revised: *“Taken together, these results indicate that in the 3D model, in which the alveolar scaffold functions as a supportive structure for attaching cells, negative pressure does not act as a stressor;”*

Effects of RM in the 3D model

- Original: *“Integrin $\beta 1$ is a well-known mechanosensor responsible for ECM adhesion and mechanical signal transduction.”*

- Revised: *“Integrin $\beta 1$ is a well-characterised mechanosensor that mediates cell adhesion by binding to components such as fibronectin, collagen, and laminin, thereby converting mechanical stimuli into intracellular biochemical responses”*

Characteristics of the ex vivo 3D lung cancer model and comparison with other models

- Original: *“However, Matrigel composition differs from that of the ECM found in the tumour microenvironment, making it challenging to truly replicate specific tissue- or cell-type-derived ECM components.”*

- Revised: *“However, the composition of Matrigel differs from that found in the tumour microenvironment (e.g., complexes containing structural proteins such as collagen and elastin), making it difficult to faithfully reproduce ECM components derived from specific tissues or cell types.”*

- Original: *“In contrast, our ex vivo 3D model retains the ECM architecture and microenvironment of actual lung tissue, allowing for the observation of cellular adhesion, proliferation, and tissue-level responses.”*

- Revised: *“In contrast, our ex vivo 3D model preserves the microenvironment of actual lung tissue, enabling observation of cellular adhesion, proliferation, and tissue-level responses.”*

• Original: *“This model can facilitate studies on ECM-mediated signalling changes that are difficult to replicate using conventional 2D culture or artificial matrices such as hydrogels. Observing ECM-specific signalling pathways—particularly those unique to lung tissue—is a key advantage of our model. However, we did not directly compare it with other widely adopted 3D models, such as organoids or ‘Lung-on-a-chip’ systems, so the relative superiority of our 3D model within the broader landscape of 3D culture systems remains to be determined.”*

• Revised: *“This model allows investigation of changes in signalling pathways dependent on the ECM, which functions as a structural and biochemical scaffold—features that are difficult to reproduce using conventional 2D culture or artificial matrices such as hydrogels. However, in the present study, we did not directly compare our model with other widely used 3D systems, such as organoids or “Lung-on-a-chip,” and therefore its relative superiority within the broader landscape of 3D culture systems remains to be determined.”*

These revisions improve readability and provide better contextualisation of the term “ECM” throughout the manuscript.

- In the References, several journal names are inconsistently italicized or abbreviated (e.g., Sci. Rep. vs Scientific Reports).

Response:

We thank the reviewer for pointing this out. We have carefully reviewed the reference list and standardised the formatting, ensuring consistent use of journal name abbreviations and italicisation throughout.

- The legend for Figure 2 does not define abbreviations like BPU and PSU upon first appearance.

Response:

We thank the reviewer for this comment. At the end of the figure legend (Figure 2), we have added the following sentence to clearly define the abbreviations: “BPU, blood pressure unit; PSU, pressure stimulation unit.” To ensure consistency with the other figures, we have adopted the style of listing all abbreviations at the end of each figure legend.

- The phrase “biologically relevant” is used repeatedly without clarification—consider substituting with more precise terminology.

Response:

We thank the reviewer for this helpful comment. We agree that the repeated use of the phrase “biologically relevant” was vague. To improve precision and clarity, we revised the text in several places, as follows:

Abstract

- Original: *“Our findings demonstrate that dimensionality and mechanical stress synergistically affect lung cancer cell dynamics and underscore the need for physiologically relevant 3D models incorporating mechanical cues for accurate cancer research.”*
- Revised: *“Our findings demonstrate that dimensionality and mechanical stress synergistically influence lung cancer cell dynamics and underscore the need for 3D models in cancer research that closely replicate the native lung tissue microenvironment.”*

Discussion – Bioreactor Mimicking a Physiological Environment

- Original: *“Our bioreactor was equipped with a pressure chamber and perfusion system, and was designed to provide an environment conducive to cell growth while applying physiologically relevant mechanical stresses.”*
- Revised: *“In our study, a bioreactor equipped with a pressure chamber and perfusion system successfully*

provided an environment in which both vascular and intrathoracic pressures remained within the physiological ranges reported in humans”

Discussion – Comparison Between 2D and 3D Models

• Original: *“In this study, we conducted a direct comparison and found that 3D models indeed provide a more physiologically relevant context, supporting their utility in lung cancer research.”*

• Revised: *“In this study, we performed such a direct comparison and confirmed that 3D models faithfully reproduce both the lung tissue microenvironment and the mechanical cues present in vivo, thereby highlighting the profound influence of model choice on cellular behaviour.”*

These changes provide clearer and more specific descriptions, thereby avoiding overuse of the vague expression “biologically relevant.”